

# Joint retrieval of surface reflectance and aerosol properties with continuous variation of the state variables in the solution space: Part 2: Application to geostationary and polar orbiting satellite observations

Marta Luffarelli[1] and Yves Govaerts[1]

[1]Rayference, Brussels, Belgium

*Correspondence to:* Marta Luffarelli (marta.luffarelli@rayference.eu)

**Abstract.** This paper presents the Aerosol Optical Thickness and surface properties simultaneous retrieval from the CISAR algorithm applied both to geostationary and polar orbiting satellite observations. The theoretical concepts of the CISAR algorithm have been described in Govaerts and Luffarelli (2017). This paper aims to demonstrate CISAR applicability to actual satellite data ac-

quired from different sensors flying on different orbits. For that purpose, CISAR has been applied to SEVIRI and PROBA-V observations acquired over 20 AERONET stations during year 2015. CISAR retrieval from the two instrument observations is evaluated against independent datasets such as MODIS land product and AERONET data. The performance differences resulting from the two types of orbit are discussed, analysing and comparing the information content of SEVIRI and

PROBA-V observations.

## 1  Introduction

Aerosol property retrieval over land surfaces from space observation is a challenging problem due to the strong radiative coupling between atmospheric and surface radiative processes. Different approaches are usually exploited to retrieve different Earth system components (*e.g.*, Hsu et al. (2013),

Mei et al. (2017)), leading to inconsistent and less accurate datasets. However, data assimilation makes use of physical models, that implicitly require input data consistency. The joint retrieval of surface reflectance and aerosol properties, as originally proposed by Pinty et al. (2000), presents many advantages, such as the possibility to perform the retrieval over any type of surface and assure the radiative consistency between the retrieved variables.





Govaerts and Luffarelli (2017) (hereafter referred to as Part I) describes the theoretical aspects of the Combined Inversion of Surface and AeRosols (CISAR) algorithm, designed for the joint retrieval of surface reflectance and aerosol properties. This new generic retrieval method specifically addresses issues related to the continuous variation of the state variables in the solution space within an Optimal Estimation (OE) framework. Through a set of experiments, CISAR capability of retriev-

ing surface reflectance and aerosol properties within the solution space was illustrated. Nonetheless, these experiments only represent ideal simulated observation conditions, *i.e.*, noise free data acquired in narrow spectral bands placed in the principal plane assuming unbiased surface prior information. This second part aims to demonstrate CISAR applicability to actual satellite observations, *i.e.*, with less favourable geometrical conditions than the principal plane and accounting for the radiometric

noise. For this purpose, the algorithm has been applied to two radiometers with similar spectral properties but different orbits, *i.e.*, geostationary and polar . Radiometers on board of geostationary platforms deliver observations with a revisit time in the range of several tenth of minutes but with a limited field of regard so that many different instruments with different sub-satellite locations are needed to cover the entire Earth. Poles cannot be observed. Conversely, a polar orbit, combined

with an adequate swath, could offer a daily revisit time of the entire globe. The selected radiometers are the Spinning Enhanced Visible and Infrared Imager (SEVIRI), flying on board of the Meteosat Second Generation (MSG) geostationary platform, and the Project for On-Board Autonomy - Vegetation (PROBA-V). These two instruments have pretty similar radiometric performances and both have acquired more than 15 years of observations thanks to the launch of a succession of radiometers

with very similar characteristics. Applying the same algorithm on similar instruments but flying on different orbits represents a meaningful way to analyse the CISAR generic algorithm performance when applied on actual data.

This paper is organised as follows. Section 2 describes the observation system considered in the OE framework, *i.e.*, the satellite observation, the ancillary information, the prior information and

the forward model. The capability of the latter to correctly simulate satellite observations is evaluated, this being one of the fundamental OE method prerequisites. The uncertainty characterisation of the observation system is also described in Section 2. The algorithm implementation is described in Section 3. Section 4 analyses the information content of the satellite observations, comparing the differences between geostationary and polar orbiting instruments, and discusses the challenges

encountered when little or no information about the retrieved variables is carried by the observation. Given these difficulties in the retrieval, a Quality Indicator (QI) is implemented and presented in Section 5, characterising the reliability of the solution. Finally, CISAR performances are discussed in detail in Section 6. CISAR retrieved Aerosol Optical Thickness (AOT) and Bidirectional Hemispherical Reflectance (BHR) will be compared against Aerosol Robotic Network (AERONET)

(Giles et al., 2017) and the Moderate Resolution Imaging Spectroradiometer (MODIS) Land product data (DAAC, 2017) respectively. The performance differences between the two retrieved datasets




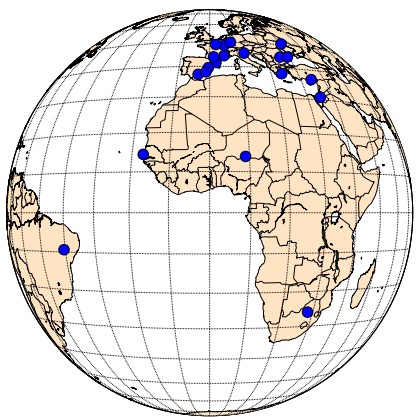

Fig. 1: Selected AERONET stations location. All stations are located within the SEVIRI field of regard.

will be further investigated through statistics on the quality of the retrieval and the information content of the satellite observations.

## 2 Observations system characterisation

### 2.1 Definition


The fundamental principle of the OE is to maximise the probability $P = P(\mathbf{x}|\mathbf{y}_{\Omega\tilde{\Lambda}}, \mathbf{x_b}, \mathbf{b})$ with respect to the values of the state vector $\mathbf{x}$, conditional to the value of the measurements and any prior information (Rodgers, 2000). The ensemble of measurements, prior information, ancillary data and the forward model constitutes the observation system. This Section describes each component of this system for the two datasets processed in the framework of this study.


In order to evaluate the CISAR algorithm's performances when applied on observations acquired with different orbits, 20 AERONET stations located within the SEVIRI field of regard have been selected (Fig. 1, Table 1) and the corresponding observations were acquired for year 2015. These targets have been selected in order to have different geometries and different land cover types (vegetation, urban, bare areas, mixed), based on the availability of AERONET observations.


For each of these stations, satellite data have been acquired, together with ancillary information, such as the cloud mask and the model parameters, *i.e.*, all the parameters that are not retrieved by the algorithm but influence the observation. Satellite data and ancillary information are accumulated in time to form a multi-angular observation vector $\mathbf{y}_{\Omega\tilde{\Lambda}}$, in order to correctly characterise the sur-





Table 1: AERONET targets

| Name | Latitude | Longitude | Land Cover Type |
|------|----------|-----------|-----------------|
| Athens_NOA | 37.99 | 23.77 | Urban |
| Barcelona | 41.39 | 2.12 | Urban |
| Bucharest_Inoe | 44.35 | 26.03 | Mixed |
| Bure_OPE | 48.56 | 5.50 | Vegetation |
| Burjassot | 39.51 | -0.42 | Urban |
| Carpentras | 44.08 | 5.06 | Vegetation |
| Dakar | 14.39 | -16.96 | Costal |
| Gloria | 44.60 | 29.36 | Water |
| Granada | 37.16 | -3.60 | Urban |
| IMS-METU-ERDERMLI | 36.56 | 34.25 | Costal |
| Kyiv | 50.36 | 30.50 | Vegetation |
| Mainz | 49.50 | 8.30 | Mixed |
| Murcia | 38.01 | -1.17 | Vegetation |
| Paris | 48.87 | 2.33 | Urban |
| Petrolina_SONDA | -9.38 | -40.50 | Urban |
| Pretoria_CSIR-DPSS | -25.76 | 28.28 | Mixed |
| Sede_Boker | 30.85 | 34.78 | Bare Areas |
| Toulouse_MF | 43.57 | 1.37 | Urban |
| Venise | 45.31 | 12.51 | Water |
| Zinder_Airport | 13.78 | 8.99 | Bare Areas |

face anysotropy. Nevertheless, retrieving surface and aerosol properties from satellite observations is an ill posed problem (Hadamard, 1902). Consequently, assumptions on the magnitude and temporal/spectral variability of the state variables are made. The ensemble of these assumptions and their associated uncertainties constitutes the prior information.

The observation uncertainty $\sigma_o$ characterisation is one of the most critical aspect of CISAR al-
gorithm as it strongly determines the likelihood of the solution. In fact the observation uncertainty determines the observation term value of the cost function, as in Eq. 17 in Part I, thus impacting the minimization process. $\sigma_o$ is composed by the radiometric uncertainty, directly related to the radiometer characteristics, the forward model uncertainty in the observed bands and the uncertainty related to the model parameters.

**2.2 Satellite data**

SEVIRI is the main instrument of the MSG mission, which has as primary objective the observation in the near real-time of the Earth's full disk, shown in Fig. 1. SEVIRI achieves this with 12 channels, ranging from 0.6 $\mu$m to 13 $\mu$m, three of which are located in the solar spectrum and centred at 0.64 $\mu$m, 0.81 $\mu$m and 1.64 $\mu$m and are used within this study. SEVIRI observes the Earth's full disk with
a 15 minutes repeat cycle. MSG nominal position is 0° over the equator in a geostationary orbit.





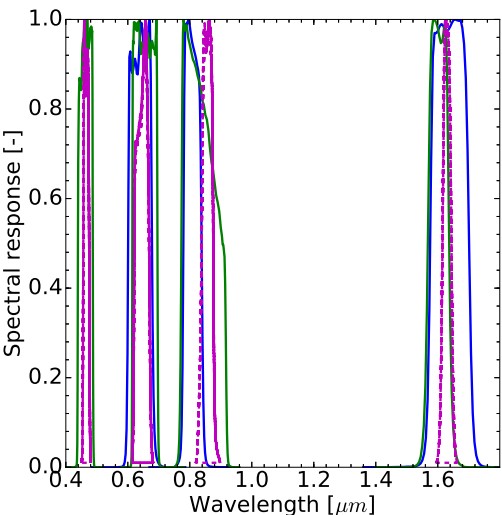

Fig. 2: SEVIRI (in blue), PROBA-V (in green) and MODIS (in magenta) spectral responses.

The sampling distance between two adjacent pixels at the sub-satellite point is 3 km for the visible bands. As there is no on-board device for the calibration of the solar channels, their calibration has been performed with the method proposed by Govaerts et al. (2013).

PROBA-V satellite mission is intended to ensure the continuation of SPOT5 VEGETATION prod-
ucts since May 2014 (Sterckx et al., 2014). The microsatellite offers global coverage of land surface with daily revisit for latitude from 75°N to 56°S in four spectral bands centred at 0.46 $\mu$m, 0.66 $\mu$m, 0.83 $\mu$m and 1.61 $\mu$m. PROBA-V products are provided at a spatial resolution of 1/3 km and 1 km, the latter being used within this study. To cover the wide angular field of view (101°) in a small-sized platform, the optical design of PROBA-V is made up of three cameras (identical three-
mirror anastigmatic (TMA) telescopes). The three cameras have an equal field of view, the central camera pointing down covers a swath of 500 km, while the right and left cameras cover 875 km each. Although the three cameras have different responses, a mean Spectral Response Functions (SRF) is considered within this study, accounting for the radiometric uncertainty associated with this approximation. Each camera has two focal planes, one for the short wave infrared (SWIR) band and one for the visible and near-infrared (VNIR) bands. Despite the different viewing angles in the SWIR band, CISAR assumes the observations are acquired with the same geometry in all bands. This assumption leads to an additional term in the observation uncertainty. Because of the omission of on-board calibration devices, the PROBA-V in-flight calibration relies only on vicarious methods (Sterckx et al., 2013).

The similarities between the three SEVIRI solar bands and the red, NIR and SWIR PROBA-V bands permit the evaluation and comparison of CISAR performances when applied to the two instru-





Table 2: PROBA-V instrument noise [%]

| **Band** | Left camera | Center Camera | Right Camera |
|------|------|------|------|
| BLUE | 4 | 4 | 4 |
| RED | 3 | 3 | 3 |
| NIR | 3 | 3 | 3 |
| SWIR | 5 | 4 | 5 |

Table 3: Total radiometric uncertainty median values [%]

|  | 0.4 $\mu$m | 0.6 $\mu$m | 0.8 $\mu$m | 1.6 $\mu$m |
|------|------|------|------|------|
| SEVIRI |  | 2.73 | 2.24 | 3.12 |
| PROBA-V | 4.01 | 3.04 | 3.02 | 4.03 |

ments, which spectral responses are shown in Fig. 2. The satellite observations have been acquired
from the European Organisation for the Exploitation of Meteorological Satellites (EUMETSAT)
Earth Observation Portal and from the Flemish Institute for Technological Research (VITO) for SE-

115    VIRI and PROBA-V respectively. The Top Of Atmosphere Bidirectional Reflectance Factor (TOA
BRF) is computed directly from the digital count value in case of SEVIRI, wheares for PROBA-V
the Level 2-A TOA BRF is delivered by VITO (Wolters et al., 2018). The satellite observation un-
certainty is derived from the radiometric noise $\sigma_i$, the geolocation uncertainty $\sigma_r$. For PROBA-V
two additional terms are calculated: the uncertainty $\sigma_c$ associated to the approximation of a mean

120    SRF of the cameras and the one deriving from considering the same viewing geometry in the SWIR
and in the VNIR bands, $\sigma_\theta$.

     PROBA-V radiometric noise has been delivered by VITO (Sindy Sterckx, personal communi-
cation, September 2017) per camera and per band Table 2. For SEVIRI, this term is computed
considering (*i*) the instrument noise due to the dark current, (*ii*) the difference between the detectors
gain and (*iii*) the number of digitalization levels (Govaerts and Lattanzio, 2007). The geolocation
uncertainty $\sigma_r$, arising from the assumption of observing always the same scene, is estimated for
each pixel **p** as follows (Govaerts et al., 2010):

$$\sigma_r^2(t,\tilde{\lambda},\mathbf{p}) = \left(\frac{\partial \mathbf{y}_0(t,\tilde{\lambda},p_x,p_y)}{\partial p_x}\sigma_x(t,\tilde{\lambda})\right)^2 + \left(\frac{\partial \mathbf{y}_0(t,\tilde{\lambda},p_x,p_y)}{\partial p_y}\sigma_y(t,\tilde{\lambda})\right)^2 \qquad (1)$$

where $\sigma_{x,y}$ is the geolocation/coregistration standard deviation and $\mathbf{y}_0(t,\tilde{\lambda},p_x,p_y)$ is the TOA BRF
in the channel $\tilde{\lambda}$ acquired at the time $t$.

     The uncertainty $\sigma_c$, originating from the usage of a mean SRF for the three PROVA-V cameras,

125    has been estimated simulating the TOA BRF considering both the mean and actual SRF for a wide
range of observation conditions. The assessed $\sigma_c$ results lower than 0.2% in all bands and for all

Table 4: Water Vapour transmittance in the SEVIRI, PROBA-V and MODIS bands

|         | 0.4 $\mu$m | 0.6 $\mu$m | 0.8 $\mu$m | 1.6 $\mu$m |
|---------|-------|-------|-------|-------|
| SEVIRI  |       | 0.993 | 0.915 | 0.988 |
| PROBA-V | 1.000 | 0.990 | 0.926 | 0.995 |
| MODIS   | 1.000 | 0.990 | 0.985 | 0.996 |

cameras. Finally, the assumption of same viewing geometry for the three PROBA-V bands is associated to the uncertainty $\sigma_\theta$, computed as follows:

$$\sigma_\Omega^2(t,\tilde{\lambda},\Omega,\mathbf{p}) = \left( \frac{\partial \mathbf{y}_0(t,\tilde{\lambda},\theta)}{\partial \theta} \sigma_\theta^2(t,\tilde{\lambda}) \right) \tag{2}$$

The total relative radiometric uncertainty median values are shown in Table 3.

### 2.3 Ancillary data

In addition to satellite observations, a cloud mask and the model parameters information are required. For SEVIRI observations, the nowcasting Satellite Application Facility (SAF) cloud mask (Meteo France, 2013), provided at the radiometer's native temporal and spatial resolution, is used; for PROBA-V the cloud mask is provided by VITO (Wolters et al., 2018). The model parameters, *i.e.*, Total Column Water Vapor (TCWV), Total Column Ozone (TCO3) and surface pressure are taken from the European Centre for Medium-Range Weather Forecasts (ECMWF) reanalysis (Dee et al.).

The uncertainties of the equivalent model parameters **b** are converted into an equivalent noise $\sigma_B$, calculated as follow (Govaerts et al., 2010):

$$\sigma_F^2(\mathbf{b},\tilde{\lambda},\Omega_0,\Omega_v) = \left( \frac{\partial y(\mathbf{x},U_{oz};\Omega,\tilde{\lambda})}{\partial U_{oz}} \sigma_{U_{oz}} \right)^2 + \left( \frac{\partial y(\mathbf{x},U_{wv};\Omega,\tilde{\lambda})}{\partial U_{wv}} \sigma_{U_{wv}} \right)^2 + \left( \frac{\partial y(\mathbf{x},U_{sp};\Omega,\tilde{\lambda})}{\partial U_{sp}} \sigma_{U_{sp}} \right)^2 \tag{3}$$

where $U_{oz}$, $U_{wv}$ are the ozone and water vapour total column concentration and $U_{sp}$ is the surface pressure and $\sigma_{U_{oz}}$, $\sigma_{U_{wv}}$ and $\sigma_{U_{sp}}$ are their associated uncertainties. The TCWV is distributed among the two atmospheric layers in the forward radiative transfer model assuming a US76 water vapour vertical profile. The fraction of TCWV in the scattering layer interacts with the aerosol particles and thus strongly affect CISAR retrieval. Unlike ozone which is mainly present in the stratosphere, water vapour is dominant in the lower part of the atmosphere, severely impacting aerosol retrieval in SEVIRI and PROBA-V band 0.8$\mu$m (Table 4). Hence, only the uncertainty related to the TCWV is considered and Eq. 3 is approximated to:





Table 5: Total EQMPN median values [%]

|          | 0.4 $\mu$m | 0.6 $\mu$m | 0.8 $\mu$m | 1.6 $\mu$m |
|----------|--------|--------|--------|--------|
| SEVIRI   |        | 0.28   | 2.02   | 0.38   |
| PROBA-V  | 0.01   | 0.37   | 1.49   | 0.14   |

$$\sigma_F^2(\mathbf{b},\tilde{\lambda},\Omega_0,\Omega_v) \approx \left( \frac{\partial y(\mathbf{x},U_{wv};\Omega,\tilde{\lambda})}{\partial U_{wv}} \sigma_{U_{wv}} \right)^2 \tag{4}$$

The median values of the Equivalent Model Parameter Noise (EQMPN), computed as in Eq. 4, are shown in Table 5.

## 2.4 Prior information

Within an OE framework, the definition of the prior information and its uncertainty plays a fundamental role. In CISAR four different sources of prior information are considered:

1. Surface parameters magnitude. The surface reflectance, represented by the RPV (Rahman-Pinty-Verstraete) model (Rahman et al., 1993), is not supposed to undergo rapid variation on a short temporal scale, hence the retrieval in the previous accumulation period can be used as prior information for the next inversion (Govaerts et al., 2010). Therefore the prior information on the RPV parameters at the time $t_d$ is built computing a running mean over the $N_r$ previous successful accumulation periods.

$$\mathbf{x}_b(t_d) = \frac{\Sigma_{t_i=0}^{t_d-1}\hat{\mathbf{x}}(t_i)}{N_r} \tag{5}$$

The corresponding prior uncertainty is defined as half of variability range of the solution $\hat{\mathbf{x}}(t_i)$ retrieved during the considered $N_r$ accumulation periods.

$$\sigma_{\mathbf{x}_b}(t_d) = \frac{\max_{t \in N_r}\hat{\mathbf{x}}(t_i) - \min_{t \in N_r}\hat{\mathbf{x}}(t_i)}{2} \tag{6}$$

When $N_r$ is smaller than a certain minimum required $N_{min}$ the prior information on the magnitude of the RPV parameters is taken from the last successful retrieval and its uncertainty is computed as in Eq. 7, where $N_d$ is the number of days since the last successful retrieval (Govaerts et al., 2017).

$$\sigma_{\mathbf{x}_b}(t_d) = \sigma_{\mathbf{x}_b}(t_d-1)1.05^{N_d} \tag{7}$$

2. AOT magnitude: this information is taken from an annual mean climatology dataset (Kinne et al., 2013). From this dataset the prior information on the AOT magnitude for the coarse and



fine mode (absorbing and non absorbing) is taken. The uncertainty is set to a high arbitrary value $\sigma_{\mathbf{x}_b}$.

3. Constraints on the AOT temporal variability. These constraints result from the assumption that the AOT is not changing rapidly on a very short temporal scale, therefore a maximum temporal variation is defined through a sigmoide function. The temporal constraints are described by the matrix $\mathbf{H}_a$ in Eq. 13 of Part I.

4. Constraints on the AOT spectral variability. The AOT is expected to decrease with the wave-
length, according to Eq. 15 of Part I. The applied constraints define the matrix $\mathbf{H}_l$ as in Eq. 14 of Part I.

## 2.5    Forward model

FASTRE, the CISAR forward Radiative Transfer Model (RTM), and its uncertainty $\sigma_F$ are described in Section 4.4 of Part I. The forward model uncertainty has been estimated for SEVIRI and PROBA-
V processed bands (Table 6). The OE method relies on the assumption that the forward model used in the inversion process is capable of correctly representing the observation. The verification of this assumption is important but difficult to realise in practice. To evaluate the validity of this assumption, a simulated dataset has been prepared with FASTRE and compared with the actual observations acquired by the two satellites in 2015 over the selected stations. To simulate the satellite
observations, FASTRE requires to know the value of the state variable (RPV parameters, AOT) and model parameters (TCWV, TCO3, Surface Pressure). The latter are taken from ECMWF reanalysis. The RPV parameters are derived from the MODIS Land Product MCD43A Collection 5 (Schaaf and Wang, 2015). The MODIS product delivers the RossLi parameters (Li and Strahler (1992), Ross (1981), Wanner et al. (1995)), from which the surface BRF can be computed. The RPV model has
been inverted against this calculated BRF field to retrieve the corresponding RPV parameters. The AOT is derived from AERONET V3 L2.0 (Giles et al., 2017), present only in clear sky conditions. The results of this evaluation are shown in Table 7. The correlation between the simulated data and actual observations exceed 0.9 in all spectral bands with a Root Mean Square Error (RMSE) that does not exceed 0.06. The relative bias exhibits values that can exceed 10%. Several factors can
explain these large values.

The spectral responses of SEVIRI, PROBA-V and MODIS are shown in Fig. 2. The different band

Table 6: FASTRE relative uncertainty in the SEVIRI and PROBA-V processed bands [%]

|  | 0.4 $\mu$m | 0.6 $\mu$m | 0.8 $\mu$m | 1.6 $\mu$m |
| --- | --- | --- | --- | --- |
| SEVIRI |  | 1.88 | 2.75 | 0.96 |
| PROBA-V | 2.38 | 1.31 | 2.20 | 0.75 |



Table 7: Comparison between FASTRE simulations and the actual SEVIRI and PROBA-V TOA BRF observations

|  | SEVIRI | | | PROBA-V | | | |
|---|---|---|---|---|---|---|---|
|  | VIS0.6 | VIS0.8 | NIR1.6 | BLUE | RED | NIR | SWIR |
| Correlation | 0.96 | 0.93 | 0.95 | 0.91 | 0.98 | 0.96 | 0.98 |
| Root Mean Square Error | 0.03 | 0.06 | 0.06 | 0.02 | 0.03 | 0.04 | 0.03 |
| Relative Bias [%] | 11.02 | 17.66 | 8.55 | 6.42 | 16.65 | 8.54 | 6.01 |

widths and center wavelengths result in differences in the gaseous absorption, as shown in Table 4. The latter is particularly important for the water vapour, as it interacts with the aerosol particles in the FASTRE scattering layer. As it can be seen from Table 4, the water vapour absorbance strongly

affects the bands centred around 0.8 $\mu$m, thus impacting the signal acquired by the three different radiometers. The different spectral responses of the three instruments, and the associated gaseous absorption, represents thus a limitation to the FASTRE model evaluation against actual observation. Furthermore, although the MODIS product provides a measure of the surface reflectance, its true value remains unknown. The lack of accurate characterisation of the surface BRF in the SEVIRI and

PROBA-V spectral bands represents also a limitation to this first FASTRE model direct comparison against actual observations. This specific point would require additional effort with dedicated ground observations. The AOT observed by the satellites could also differ from the AERONET one, given the larger spatial resolution of the satellite observations, which could results in cirrus contamination or neighbouring aerosol events not caught by the AERONET measure. Given these considerations,

although FASTRE simulated satellite observations with a correlation of about 0.9 and in a first approximation results suitable for inversion purposes, more effort would be needed to demonstrate that the forward RTM is unbiased.

## 3   Data processing

### 3.1   General setup

The theoretical concepts of the CISAR algorithm have been described in Part 1. In order to perform the inversion on actual satellite data, the observations are accumulated in time and the corresponding uncertainty is computed as described in Section 2. This temporal accumulation is performed in order to build a multi-angular observation vector $\mathbf{y}_{\Omega\tilde{\Lambda}}$ to characterise surface anisotropy. The surface optical properties are considered invariant during the accumulation period, hence a trade-off between

having enough cloud free observations to build the observation vector and allowing the algorithm to catch surface variations is introduced; the high repeat coverage of geostationary satellites allows a shorter accumulation periods with respect to polar orbiting instruments. For SEVIRI acquisitions,



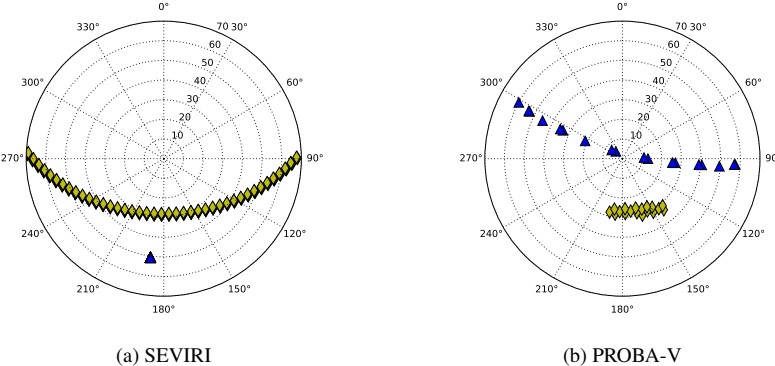

|          (a) SEVIRI          |          (b) PROBA-V          |

Fig. 3: Polar plot of the angular sampling during a 5 days (2015/05/01-2015/05/05) of SEVIRI observations (left panel) and during 16 days (2015/05/01-2015/05/16) of PROBA-V observations (right panel) over Carpentras, France. The blue triangles represent the viewing geometry, the yellow diamonds the illumination one. Circles represent the zenith angle and polar angles represent azimuth angles with zero azimuth pointing to the North.

although the angular sampling does not vary much from one day to the next, the length of the accumulation period is set to 5 days in order to maximise the occurrence of cloud free observations.

For polar orbiting satellite the length of the temporal accumulation is normally driven by the repeat cycle, as it is done for MODIS observations (DAAC, 2018). In the case of PROBA-V, the satellite orbit is not maintained, *i.e.*, there is no repeat cycle. Hence, the choice of the length of the time window during which the satellite observations are accumulated results from empirical studies aim to balance the trade-off previously described. Consequently, the length of the accumulation is set to

16 days and the successive accumulation periods are shifted by 8 days. An example of the angular sampling during this accumulation period is shown in Fig. 3 for SEVIRI and PROBA-V. During the accumulation process, observations acquired with a sun and viewing angle larger than $\theta_{max}$ (defined in Table 8) are discarded.

Table 8: CISAR setup parameters

|                |                                                      | SEVIRI | PROBA-V |
|----------------|------------------------------------------------------|--------|---------|
| $N_d$          | Length of the accumulation period                    | 5      | 16      |
| $N_s$          | Shift between the accumulation period                | 5      | 8       |
| $\theta_{max}$ | Maximum processed sun and viewing zenith angles [°]  | 70     | 70      |
| $\tau_{low}$   | Minimum AOT first guess value                        | 0.001  | 0.001   |
| $\tau_{high}$  | Maximum AOT first guess value                        | 0.100  | 0.100   |
| $\sigma_{x_b,\tau_F}$ | Fine mode prior uncertainty for the AOT       | 1.0    | 1.0     |
| $\sigma_{x_b,\tau_C}$ | Coarse mode prior uncertainty for the AOT     | 2.0    | 2.0     |
| $\sigma_{x_b,RPV}$    | Default prior uncertainty for the RPV parameters | 1.0  | 1.0     |





At the end of this accumulation period the inversion takes place. The definition of the first guess is an important aspect of the inversion process and it is defined in order to minimise the possibility of finding local minima. When a minimum value is found, an exploration should be made in order to determine whether or not it is a local minimum. However, this exploration could be computationally expensive. In order to minimise the possibility of local minima without degrading the computational performances, the AOT first guess is set alternating between a low value $\tau_{low}$ and a larger one $\tau_{high}$ (see Table 8). As CISAR retrieves one single set of RPV parameters over the entire period in each processed band, only one set of first guesses $\mathbf{x}_0$ is defined:

$$\mathbf{x}_0(t_d) = \mathbf{x}_b(t_d) + (-1)^{i_{t_d}} * \sigma_{\mathbf{x}_b}(t_d) \tag{8}$$

where $i_{t_d}$ is the index of the current accumulation period and $\mathbf{x}_b$ is the prior information at the accumulation period $t_d$.

From the retrieved set of RPV parameters the BHR is calculated, assuming perfectly diffuse illumination conditions, and the AOT is extrapolated at 0.55 $\mu$m through the extinction coefficient $\alpha$:

$$\tau_{0.55,v} = \tau_{\lambda,v} \left( \frac{\alpha_{0.55,v}}{\alpha_{\lambda},v} \right) \tag{9}$$

where $v$ is the considered aerosol vertex and $\lambda$ is the wavelength from which the AOT at 0.55 $\mu$m is extrapolated. The setup parameters are summarized in Table 8.

## 3.2 Aerosol vertices

The choice of the aerosol vertices determines the solution space in which the aerosol properties can be retrieved. The relationship between the particle size and the single scattering properties has been discussed in Part I. As recommended in the latter, three vertices are selected, defined by the asymmetry factor $g$ and the Single Scattering Albedo (SSA) $\omega_0$: two fine mode vertices, absorbing and non-absorbing, and one coarse mode vertex, defining a triangle in the $[g,\omega_0]$ space in each processed band. The three vertices are chosen analysing the single scattering properties derived from the AERONET inversion product on all available observations since 1993 (Dubovik et al., 2006), similarly to the approach proposed by Govaerts et al. (2010). The aerosol single scattering properties distribution in the $[g,\omega_0]$ space, as derived from AERONET inversion product, is shown in Fig. 4 for $\lambda = 0.6$ $\mu$m. It can been seen that the aerosol properties are clustered in the region defined by $0.60 < g < 0.80$ and $0.85 < \omega_0 < 0.98$, containing 68.3% of the data (blue line). The selected CISAR vertices defining the solution space cover about the 90% of possible solutions (magenta line).



## 4 Information content

The analysis of the information content relies on a two-fold approach. First, the Jacobians are used as an indicator of the TOA BRF sensitivity to state variable changes under different observation

conditions. Next, the entropy is used as a rigorous metric to determine the information content of the observation system for each radiometer. The Jacobians, *i.e.*, the partial derivatives of the forward model with respect to the state variables, are affected by the changes in illumination and viewing geometry both in terms of sign and magnitude (Luffarelli et al., 2016). Representing the sensitivity of the TOA BRF on the state variables, the Jacobians are a key parameter in the inversion

process. The latter consists in fact in the minimisation of the cost function, defined by Eq. 16 of Part I. This minimisation relies on an iterative approach where the descent direction is determined by the Jacobians (Marquardt, 1963). An intuitive analysis of the Jacobians gives a first insight into the amount of information carried by the observation and the challenges associated to its variations throughout the year, both in term of sign and magnitude. The higher the magnitude of the Jacobians,

the higher the sensitivity of the signal on the selected state variable.

An example of the distributions of the Jacobians related to the RPV parameters is shown in Fig. 5 (Carpentras, France). It can be seen that the Jacobians are dominated by the $\rho_0$ parameter (controlling the magnitude of the surface BRF), followed by $\theta$, $k$ and $\rho_c$ (characterising the surface anisotropy). Consequently, the retrieval of the surface reflectance shape results more challenging

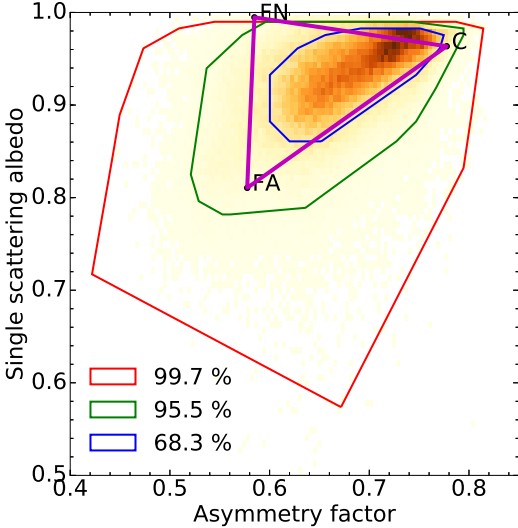

Fig. 4: Solution space for the wavelength $0.6\mu$m defined by the non absorbing fine mode (FN), the absorbing fine mode (FA) and the coarse mode (C) vertices. The red, green and blue lines show respectively the 99.7%, 95% and 68% probability regions respectively, as derived from AERONET inversion product for all the observations available over all the AERONET stations.




with respect to its mean magnitude; nevertheless, its accurate retrieval is necessary to correctly account for the coupling between the surface and the atmosphere (Govaerts et al., 2008).

The aerosol contribution to the signal at the satellite differs according to the brightness of the surface. Figure 6 shows the AOT Jacobians distribution over Carpentras (dark surface) and Zinder Airport (bright surface). It can be seen that the Jacobians over Carpentras reach higher values with
respect to the Jacobians related to Zinder Airport, where the signal coming from the bright surface is larger with respect to dark targets (Sun et al., 2016). When magnitude of the AOT Jacobian is close to 0 the observed TOA BRF is not sensitive to changes in the aerosol concentration in the atmosphere. It is worth noticing that the aerosol Jacobians can be both negative and positive, meaning that the aerosols can increase or decrease the TOA BRF depending on the season and the
viewing and illumination geometry. The sign of the partial derivatives describes in fact in which way the state variables contribute to the signal. For instance, if the partial derivative of the TOA BRF with respect to the AOT is positive (negative), an increase in the aerosol concentration will increase (decrease) the signal. This variability of the sign of the Jacobian, occurring also over dark target, as shown in Fig. 6a, represent one limitation in the MODIS Dense Dark Vegetatation (DDV)
algorithm (Kaufman et al., 1997), which assumes that an increase in the AOT results in an increased signal at the satellite, *i.e.*, this approach can only handle positive Jacobian values. Table 9 shows median values and standard deviation for all the state variables at SEVIRI and PROBA-V centred at 0.6 $\mu$m over all selected AERONET stations over year 2015. This Table confirms the previous findings on the Jacobians magnitude shown in Fig. 5 and 6 over Carpentras and Zinder Airport. The
variability of the Jacobian sign and magnitude along the year is illustrated in Fig. 7, where it can be seen that the effect of the aerosols on the reflectance can vary with the geometry for the same land cover type. The Jacobian variations in Fig. 7 essentially depend on the viewing and illumination

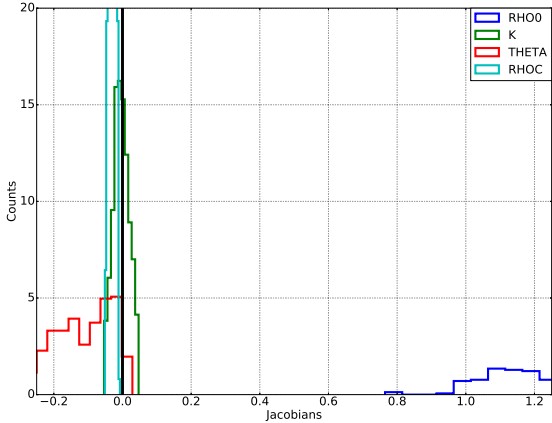

Fig. 5: Distribution of the Jacobians related to the RPV parameters. These distributions are obtained from PROBA-V observations (RED band) over Carpentras, France (vegetated target).





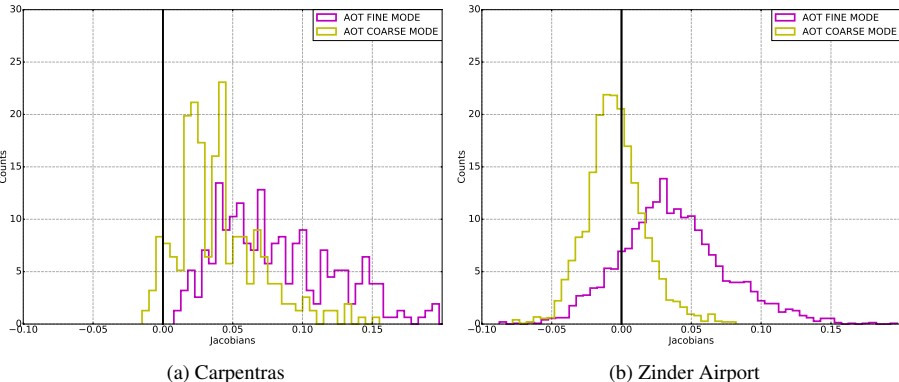

|     (a) Carpentras     |     (b) Zinder Airport     |

Fig. 6: Distribution of the AOT Jacobian over Carpentras (dark surface) and Zinder Airport (bright surface). The histograms are obtained from PROBA-V observations (RED band) over year 2015.

geometry. Aerosol particles mostly scatter in the forward direction, given the positive sign of the asymmetry factor $g$ (controlled, among other factors, by the aerosol size) (Andrews et al., 2006).

For this reason, the maximum information on the aerosols is located in the forward direction, while it decreases when approaching the backscattering direction. Additionally, a longer atmospheric path increases the aerosol effects on the reflectance, given the higher probability of interactions between the reflected sunlight and the atmospheric particles. The impact of the length of the atmospheric path is highlighted in Fig. 8, showing the Jacobian daily cycle over Carpentras. The sensitivity of

the TOA BRF with respect to the AOT almost disappears at noon, when the atmospheric path is shortest and the signal coming from the surface is dominant with respect to the aerosols. The above considerations on the Jacobian have shown that the dominant contribution to the TOA BRF comes from the $\rho_0$ parameter, controlling the surface reflectance magnitude. The retrieval of the anisotropy of the surface appears to be more challenging, given the low dependency of the TOA BRF on it.

This in turn makes the retrieval of the AOT challenging, given the radiative coupling between the atmosphere and the anisotropy of the underlying surface (Wagner et al., 2010). Also, given the

Table 9: Median and standard deviation of state variables Jacobians. The table refers to all processed targets during 2015. The values are shown for the SEVIRI and PROBA-V bands centred at 0.6 $\mu$m.

|           | Median value | Standard deviation |
|-----------|--------------|--------------------|
| $\rho_0$  | 1.410        | 0.758              |
| $\kappa$  | -0.065       | 0.079              |
| $\theta$  | -0.148       | 0.200              |
| $\rho_c$  | -0.033       | 0.031              |
| $\tau_F$  | 0.088        | 0.135              |
| $\tau_C$  | 0.035        | 0.070              |





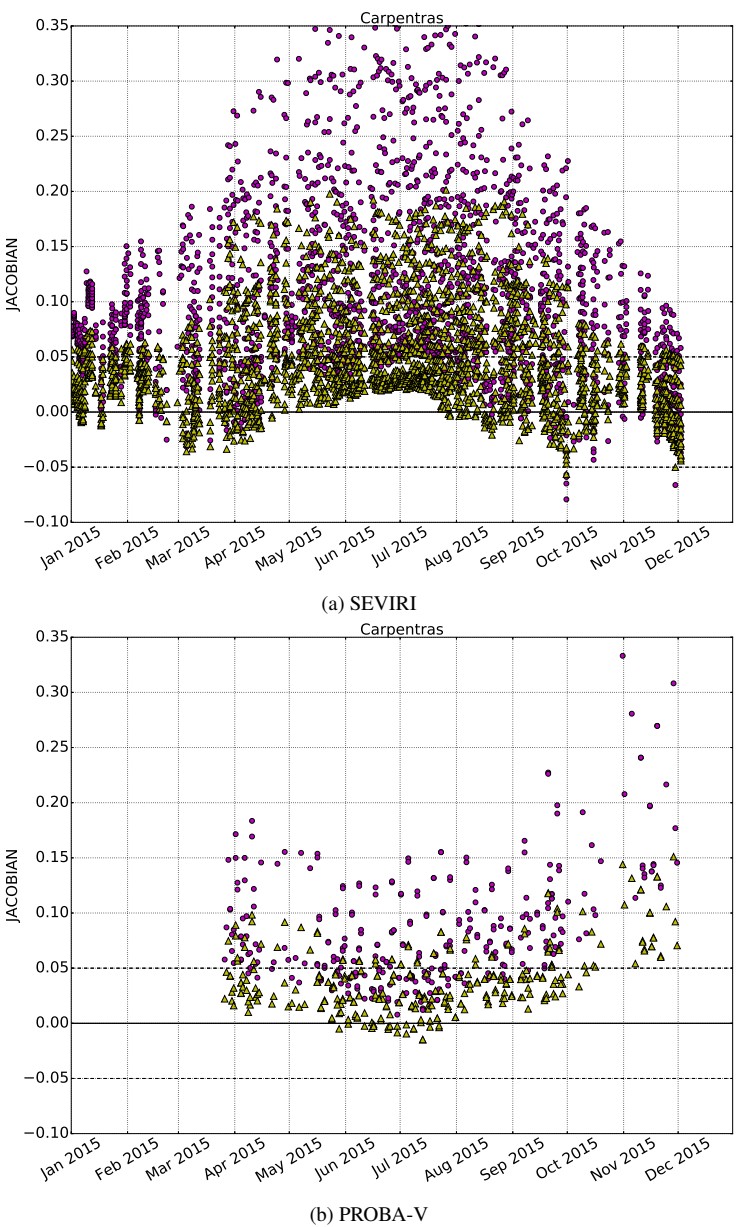

(a) SEVIRI

(b) PROBA-V

Fig. 7: AOT Jacobians timeseries over Carpentras, France (vegetated target) related to SEVIRI VIS0.6 band (top panel) and PROBA-V RED band (bottom panel) observations. The magenta dots represent the fine mode, the yellow triangles the coarse mode.



seasonal variations of the Jacobian, shown in Fig. 7 and 8, it is not easy to have the same accuracy of the retrieval throughout the day and throughout the year.

The interpretation of the Jacobians allows an intuitive understanding of the sensitivity of the TOA BRF at the satellite with respect to the state variables. On the other hand, a more rigorous analysis of the information content can be made through the entropy, *i.e.*, the measure of the uncertainty reduction (Rodgers, 2000). In an OE framework, the prior information and its uncertainty represent an hypothesis on the expected value of the state variables. It is envisaged that the inversion process provides a posterior uncertainty on the state variables which is smaller than the prior one; the entropy quantifies this uncertainty reduction. When there is no information coming from the satellite observations, the entropy will be close to 0 as the observation does not add any additional knowledge on the system. Formally, the entropy is computed as follows:

$$H = -\frac{1}{2} ln \left( \frac{|\mathbf{S}_{\hat{x}}|}{|\mathbf{S}_x|} \right) \qquad (10)$$

where $\mathbf{S}_{\hat{x}}$ is the uncertainty of the posterior (Eq. 21 of Part I) and $\mathbf{S}_x$ is the uncertainty of the prior

information.

In CISAR, the entropy is calculated for the RPV parameters and the AOT separately; its distribution is shown in Fig. 9 and 10 respectively. As it can been observed the distribution of the surface and AOT entropy related to SEVIRI observations exhibits higher values compared to the one related to PROBA-V observations, meaning a higher information content from geostationary satellites

observations with respect to polar orbiting ones, especially for the AOT. It should be noticed that the entropy depends not only on the information carried by the satellite observation, but also on the uncertainty associated to the prior information. As the prior information on the surface is updated as

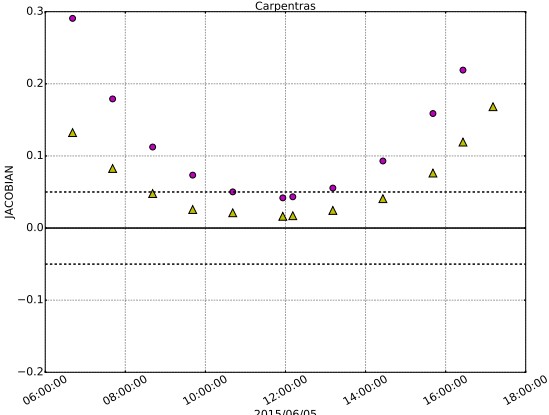

Fig. 8: AOT Jacobians associated to SEVIRI observation over Carpentras, France, for 2015/6/5. The magenta dots represent the fine mode, the yellow triangles the coarse mode.

in Section 2.4, the uncertainty associated with it decreases in time, whereas the prior information on the AOT remains weakly constrained, *i.e.*, the uncertainty is kept to the default high value. This is

the reason why the entropy associated to the RPV parameters (Fig. 10) exhibits smaller value than the one associated to the AOT (Fig. 9). Geostationary satellites acquire observations at a higher temporal frequency than polar orbiting one; the multi-angular vector obtained by accumulating satellite observations over time will thus offer a larger and more diverse angular sampling with respect to polar orbiting satellite observations (Fig. 3). This, in turn, results in a higher probability of correctly

characterising the surface anisotropy and its coupling with the atmosphere (Wagner et al., 2010).

## 5   Quality indicator

### 5.1   Principle

In Section 2.5 the OE assumption of the forward model being capable of correctly characterising the satellite observation has been discussed. However the CISAR forward model, FASTRE, described

in Part I, is a simple 1D model which is not always capable of fitting the complexity of the observations. Furthermore, in Section 4 it has been shown how the AOT Jacobian magnitude is subject to daily and seasonal variation, depending on the viewing and illumination geometry. These issues compromise the reliability of the retrieved solution, which can however be assessed using different methods. Dubovik et al. (2011) use the relative fitting measurement residual, *i.e.*, the observation

term of the cost function, to filter the retrieval outliers. Such approach presents some limitations as the probability to have a good fit by chance increases as the numbers of cloud-free observations decreases. To address this specific issue, Govaerts and Lattanzio (2007) developed an approach based on the residual of the cost function, but also taking into account the number of cloud free

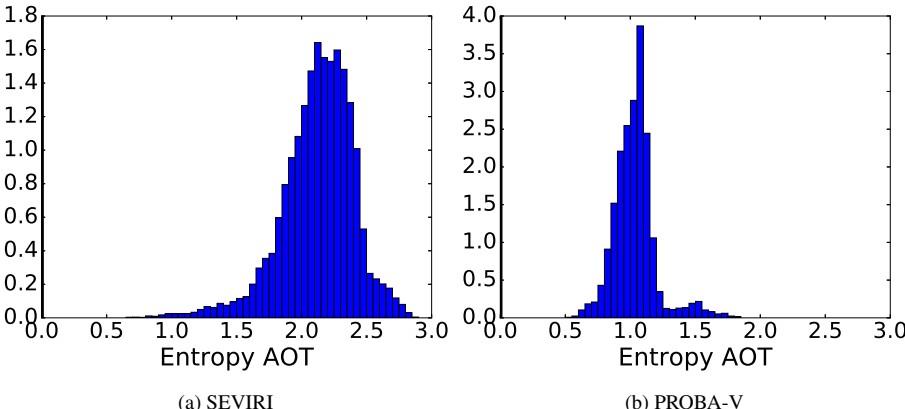

(a) SEVIRI                                    (b) PROBA-V

Fig. 9: Distribution of the entropy related to the aerosols properties for SEVIRI (left panel) and PROBA-V (right panel) respectively.





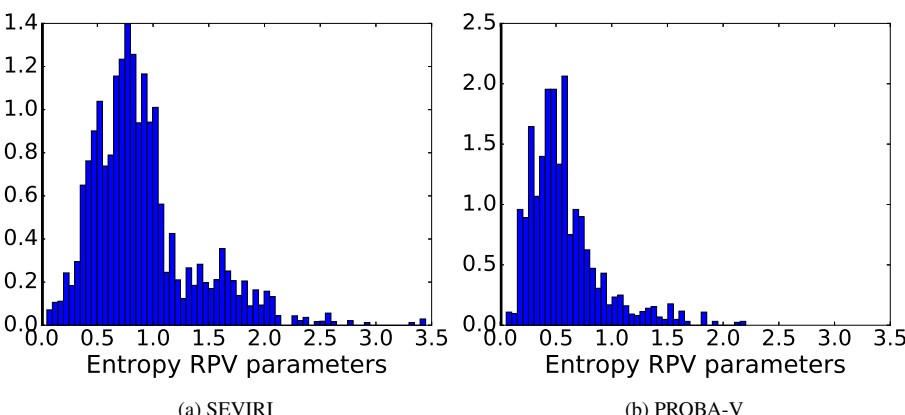

(a) SEVIRI  (b) PROBA-V

Fig. 10: Same as Figure 9 but related to the surface space.

observations. The authors observed that the latter can vary between pixels, and that the cost function

is proportional to the quadratic sum of the miss-fit between the simulation and the observation for

each acquisition, weighted by the observation uncertainty. For these reasons, as the cost function is

strongly dependent on the number of observations, it is not possible to define a universal range of

acceptable values for its residual.

Nevertheless, both methods do not correctly identify situations in which a good fit of the TOA BRF

is reached but the retrieval of the state variables is not reliable, due to limited or no dependency of

the TOA BRF on the state variables, *i.e.*, the Jacobians are close to 0. A more elaborated QI has

been developed for MODIS Aerosol Product Collection 6 (Hubanks, 2017), which is composed of

different tests accounting for the fitting residual, the magnitude of the retrieved AOT, the possible

presence of cirrus, the brightness of the scene and informations on the number of pixels and the

percentage of water pixels present in the processed area. Despite taking into account different factors

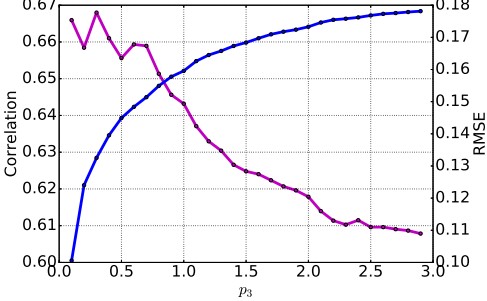

Fig. 11: Correlation (in purple) and RMSE (in blue) variations in function of the miss-fit test $p_3$.
The figure refers to the CISAR AOT retrieval evaluation against AERONET data. These results are
obtain from CISAR applied to SEVIRI observations.





in addition to fitting residuals, this approach does not consider the actual information content of the satellite observation. Moreover, as CISAR performs the retrieval on a pixel-level, the information on the number and type of pixels over which the retrieval is performed, as used in the MODIS product, is not applicable within this method. A new approach is therefore proposed here, which combines a series of individual tests $p_i$ to defined a $QI(t_i)$ associated to each retrieved solution $t_i$. The proposed method takes into account the convergence of the inversion to a solution after a given number of iterations ($p_0$), the validity range of the total AOT ($p1$) and surface albedo ($p_2$), the miss-fit between observations and simulations ($p_3$) and the information content of the satellite acquisition through the Jacobians ($p_4$) and the entropy, as discussed in Section 4. The entropy is computed separately for the AOT ($p_5$) and RPV parameters ($p_6$). These tests have been defined trough an analysis of their impact on CISAR performance when evaluated against independent datasets (see Supplement). Figure 11 shows an example of the retrieved AOT evaluation against AERONET data for the miss-fit test ($p_3$); it can been seen that a better fit leads to a smaller RMSE and a higher correlation.

## 5.2 Theoretical concept

### 5.2.1 Convergence

The first tests to be performed is on the convergence of the inversion $p_0$; when the maximum number of iteration is reached, the entire processed accumulation period is assigned a QI equal to 0. The convergence is the first parameter to be analysed because if this test fails, *i.e.*, CISAR has not actually converged to a solution, the other tests are not performed.

### 5.2.2 State variable validity range

The validity of the retrieved total AOT $p_1$ and of the surface BHR $p_2$ is investigated. In fact a validity range is defined, based on physical boundaries and empirical observations. When the retrieved state variable is equal to the minimum or maximum value determining this range, a QI=0 is assigned to the retrieval. The minimum and maximum values defining the AOT and BHR validity range are reported in Table 10. Normally, very high values of AOT (close to 5.0) indicates cloud contamination.

Table 10: Validity range for the total AOT and BHR.

|     | Min | Max |
| --- | --- | --- |
| BHR | 0.0 | 1.0 |
| AOT | 0.0 | 5.0 |





### 5.2.3 Miss-fit between observation and simulation

As discussed in the Section 5.1, the fitting residual between the observation and the simulation is normally used to assess the reliability of the solution, as it describes if the signal simulated with the forward model correctly characterise the satellite observations. The miss-fit between the observed and simulated TOA BRF is evaluated in terms of the observations uncertainty:

$$p_3(t) = \frac{\mid y_m(t) - y_o(t) \mid}{\sigma_o(t)} \tag{11}$$

where $y_m(t)$ is the BRF calculated by the forward model at time $t$, $y_o(t)$ is the observation and $\sigma_o(t)$ is the observation uncertainty. As anticipated in Section 2.1 the observation uncertainty plays a fundamental role in the minimisation problem, appearing as denominator in the cost function obser-
vation term (Eq. 17 Part I). For the inversion being successful, the difference between the simulated signal and the satellite observation should be entirely justified by the observation uncertainty $\sigma_o(t)$, *i.e.*, it should results $p_3(t) \leq 1$.

### 5.2.4 Jacobian

As discussed in Section 4, the Jacobians give information on the sensitivity of the TOA BRF on the
state variables. Performing a test over the Jacobians magnitude related to each state variable might however be computationally expensive. To reduce the computational effort, only the Jacobian of the AOT is taken into account. This choice derives from the analysis of the Jacobian distributions related to the retrieved state variables, which median values and standard deviations are reported in Table 9. It can been observed that the surface anistropy parameter $\kappa$, the fine and coarse mode
AOT and the hotspot parameter $\rho_c$ present low Jacobian magnitude with a small standard deviation. However, Maignan et al. (2003) evaluated the impact of a correct hotspot modeling on the surface albedo estimate. They found that, being limited to a small angular range, the impact of the hotspot effect on the albedo is negligible. For this reason, it has been chosen to discard the analysis of the $\rho_c$ Jacobian for the QI computation. Additionally, the surface prior update mechanism described
in Section 2.4 provides a robust estimate of the surface BRF parameters. Hence, only the AOT Jacobians are considered for the elaboration of the QI. Moreover, the spectral constraints applied on the AOT variability as in Section 2.4 impose a correlation between the AOT retrieved in the different spectral bands. Consequently, it is desirable to have information in at least one band, but this information has to be present in each aerosol vertex in order to have good retrieval of the total
AOT. Hence, the parameter considered in this quality assessment is computed as follows:

$$p_4(t) = \max_{\lambda} \left\{ \min_{v} \left\{ J_{\tau_{\lambda,v}}(t) \right\} \right\} \tag{12}$$

where $\lambda=1,\dots,$number of wavelengths and $v=1,\dots,$ number of aerosol vertices.



### 5.2.5 Entropy

Section 4 discusses the entropy computation as a rigorous analysis of the information content, quantify the uncertainty reduction from the prior knowledge on the system to the posterior uncertainty after the inversion. This test analyses separately the entropy related to the AOT ($p_5$) and the RPV parameters ($p_6$), compute as follows:

$$
\begin{aligned}
p_5(t) &= -\frac{1}{2N_\lambda} ln\left( \frac{\prod_\lambda \prod_c \sigma_{post_{\lambda,v}}}{\prod_\lambda \prod_v \sigma_{prior_{\lambda,v}}} \right) \\
p_6(t) &= -\frac{1}{2N_\lambda} ln\left( \frac{\prod_\lambda \prod_p \sigma_{post_{\lambda,p}}}{\prod_\lambda \prod_p \sigma_{prior_{\lambda,p}}} \right)
\end{aligned}
\tag{13}
$$

where $N_\lambda$ is the number of processed wavelengths, $\lambda = 1, \ldots, N_\lambda$, $p = 1, \ldots$, number of RPV parameters and $v = 1, \ldots$, number of aerosol vertices. The normalization to $N_\lambda$ assures consistency in the entropy evaluation when different number of bands are analysed, as for SEVIRI and PROBA-V cases.

### 5.3 Quality indicator computation

The final QI is computed combining the results of the tests performed on the retrieved solution. As described in Section 5.2.1 and 5.2.2 a QI equal to 0 is assigned if one test among $p_0$, $p_1$ and $p_2$ fails. For the quantities $p_i$, with $i = 3 \ldots 6$, a minimum $T_{1,i}$ and a maximum $T_{2,i}$ thresholds are defined to discriminate between good and poor quality retrievals and a intermediate indicator $q_i$ is assigned as follows:

$$
\begin{cases}
q_i = 0 & \text{if} \quad p_i(t) \leq T_{1,i} \quad \forall \quad i = 4,5,6 \quad \text{or} \quad p_i(t) \geq T_{2,i} \quad \forall \quad i = 3 \\
q_i = 1 & \text{if} \quad p_i(t) \geq T_{2,i} \quad \forall \quad i = 4,5,6 \quad \text{or} \quad p_i(t) \leq T_{1,i} \quad \forall \quad i = 3 \\
m_i < q_i < 1 & \text{if} \quad T_{1,i} \leq p_i(t) \leq T_{2,i} \quad \forall \quad i = 3,4,5,6
\end{cases}
\tag{14}
$$

where $m_i$ is the minimum value assigned to $q_i$. These weights are selected according to the principle behind the test $i$ definition and the different impact that the different quantities $p_i$ have on the fitting between the CISAR retrieval when evaluated as explained in Section 5.1. For instance, a larger weight is given to the test on the miss-fit, $q_3$, as it indicates whether the forward model was capable of correctly simulated the observation, this being one of the main assumption of the OE method as explained in Section 2.5. When $T_{1,i} \leq p_i(t) \leq T_{2,i}$ the value of $q_i$ is determined non-linearly through a sigmoide function as in Fig. 12. In fact, as the quality of the retrieval does not change rapidly moving away from the thresholds $T_{1,i}$ and $T_{2,i}$, the smoother transition obtained with a sigmoide function has been preferred to a linear transformation.

Finally, the $QI(t)$ associated with each retrieval $t$ is computed as follows:

$$
QI = p_0 p_1 p_2 \left( 1 - \max\left\{ \sum_{i=3}^{6} (1-q_i), 0 \right\} \right)
\tag{15}
$$





The selection of the thresholds $T_{1_i}$ and $T_{2,i}$ results from the *a posteriori* evaluation of the CISAR retrieval against independent datasets as in Section 5.1 and based on the meaning of each test $i$. For instance, $T_{1,3}$, *i.e.*, the minimum threshold related to the miss-fit between observed and simulated TOA BRF test, is set to 1, as $p_3 \leq 1$ means that the difference between the simulated signal and the satellite observation does not exceed the observation uncertainty $\sigma_o$. Conversely, $T_{2,3}$ has

been chosen observing the impact of $p_3$ on the fitting between CISAR retrieval and the independent datasets used as reference (Fig. 12 represents an example of this evaluation). $T_2$ has thus been set to 2.0, meaning that the maximum tolerance on the miss-fit is equal to $2\,\sigma_o$. Regarding the test on the Jacobian magnitude, its aim is to discard observations with Jacobians close to 0, *i.e.*, where there is little or no dependency of the TOA BRF on the AOT rather than associate a large Jacobian to a good

quality retrieval. In other words, this test intends to identify those situations where the test on the miss-fit is successful because of the prior information and/or the temporal and spectral constraints (Section 2.1) rather than actual information coming from the observations. The thresholds $T_{1,4}$ and $T_{2,4}$ are therefore set to 0.01 and 0.02 respectively, in order to filter out observations where the Jacobian magnitude is close to 0. Finally, the tests performed on the entropy are strongly dependent on

the magnitude of the prior uncertainty as explained in Section 4. For this reason an additional step is performed on $p_5$ and $p_6$, *i.e.*, they are only performed when the prior uncertainty is smaller than the validity range of the AOT and RPV respectively and larger than 1/6 of it. The thresholds associated to the two tests on the entropy are $T_{1,5} = T_{1,6} = 0.1$ and $T_{2,5} = T_{2,6} = 0.6$ that correspond to 20% and 70% uncertainty reduction respectively. Figure 13 shows the variations of the correlation and

the RMSE between CISAR retrieved AOT and AERONET data as a function of the QI. Correlation increases as QI is taking large values while the RMSE decreases. This behaviour is observed with CISAR AOT retrieved from both SEVIRI (Fig. 13a) and PROBA-V (Fig. 13b) observations. This correlation increase (RMSE decrease) is particularly visible when QI is taking values between 0.0

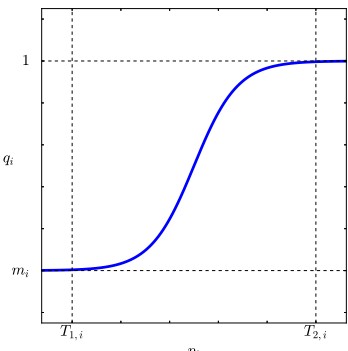

Fig. 12: Non linear $q_i$ definition between a minimum $m_i$ and 1 which applies for the parameter $p_i$ when $T_{1,i} \leq p_i(t) \leq T_{2,i}$.





and 0.2. For this reason, in Section 6 only retrievals with $QI \geq 0.2$ are considered.

## 6   Performance evaluation

### 6.1   Aerosol Optical Thickness

CISAR AOT retrieval, extrapolated at 0.55 $\mu$m has been evaluated against AERONET data over
the selected targets listed in Section 2. CISAR AOT retrieval is evaluated in terms of correlation,
RMSE, Mean Absolute Bias (MAE) with respect to AERONET values. Additionally, the percentage

of points falling within the Global Climate Observation System (GCOS) requirements (Systematic
Observation Requirements for Satellite-Based Data Products for Climate, 2011 Update), defined as
$\max\{0.0300, 10\%\}$, is also accounted for. The GCOS requirements are a useful tool to compare dif-
ferent algorithms' performances. However, it should be considered that both SEVIRI and PROBA-V
missions were not originally designed for AOT retrieval, whereas the GCOS requirements establish

the standards for the generation of Essential Climate Variables (ECVs) derived from dedicated mis-

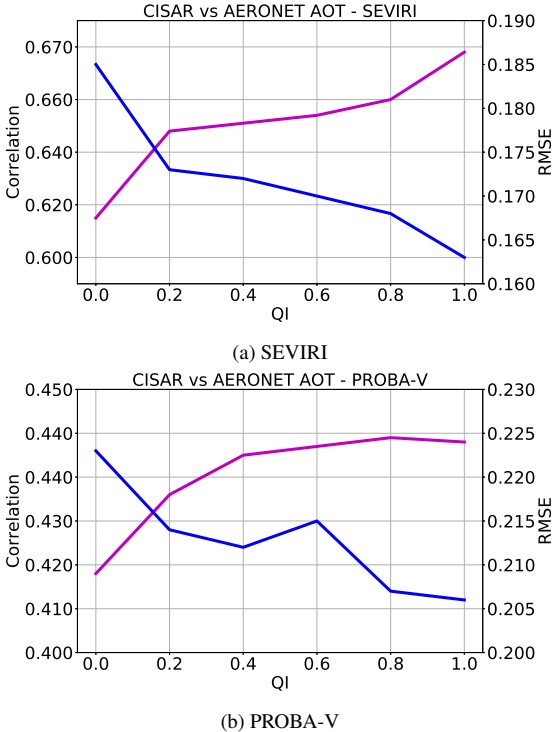

Fig. 13: Correlation (in purple) and RMSE (in blue) variations in function of the QI. The figure refers
to the CISAR AOT retrieval from SEVIRI (top panel) and PROBA-V (bottom panel) observations
evaluated against AERONET data.





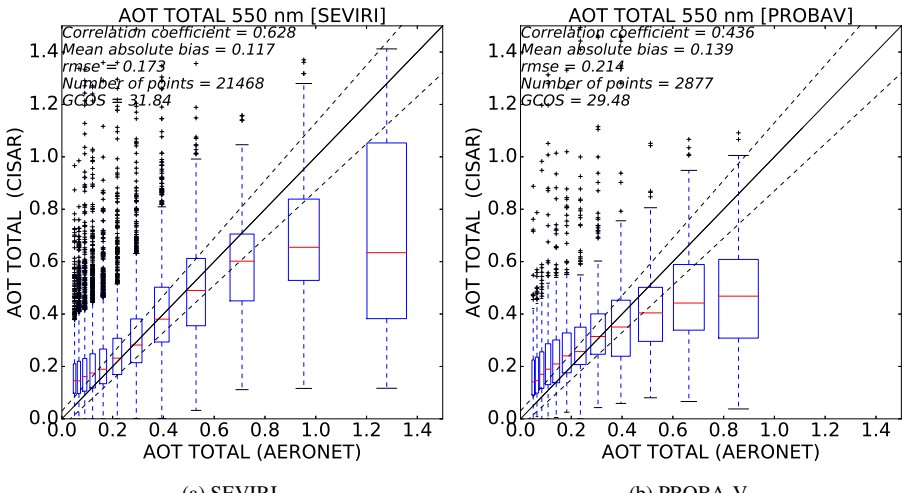

(a) SEVIRI  (b) PROBA-V

Fig. 14: Boxplot between CISAR total AOT retrieval (extrapolated at 0.55 $\mu$m) and the AERONET data for both SEVIRI (left panel) and PROBA-V (right panel) over all the selected stations in 2015. Only retrieval with $QI \geq 0.2$ are considered. The blue boxes represent the interquartile range ($IQR$), the red horizontal line inside the blue boxes represents the median value, the vertical dashed bars represent the $1.5 * IQR$ range and the black crosses represent the outliers.

sions. In the following, the GCOS requirements are evaluated in terms of percentage of retrievals satisfying them. The duration of the corresponding missions provides however a decisive advantage for the generation of AOT ECVs.

Figure 14 shows the evaluation of the retrieved AOT against AERONET data for both SEVIRI 450 (left panel) and PROBA-V (right panel). CISAR retrieval from SEVIRI observations shows a better agreement with the AERONET data compared to the retrieval from PROBA-V observations. This

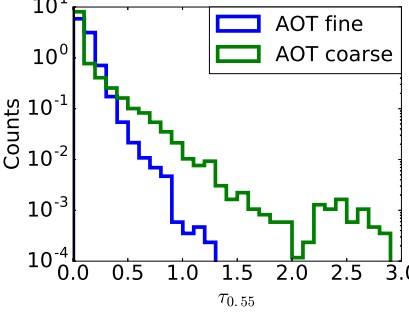

Fig. 15: Fine (blue) and coarse (green) mode at 0.55 $\mu$m distributions from AERONET observations over the selected stations (Table 1) over year 2015.

(c) Author(s) 2018. CC BY 4.0 License.



is in accordance with the outcome of the information content analysis performed in Section 4. The boxplots in Fig. 14 show a slight overestimation of the retrieval for low AOT ($\tau < 0.2$) and an underestimation for large AOT ($\tau > 0.6$). A similar behaviour is also observed in Wagner et al. (2010).

The underestimation for large values might be partially due to the temporal constraints described in Section 2.4, as they might prevent the algorithm to fit rapidly evolving aerosol events associated with large AOT values. However the applied temporal constraints are intended to optimise the retrieval of low aerosol concentration, given the global distribution of AOT which normally results smaller than 0.2 (Kokhanovsky et al., 2007). A second possible explanation is related to the low Jacobians

of the AOT, particularly the coarse mode. High AOT are in fact generally related to large particle

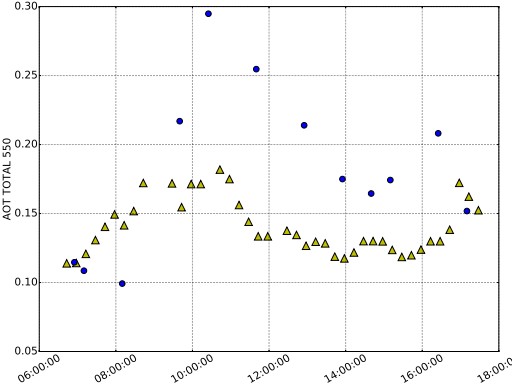

(a) CISAR (blue dots) and AERONET (yellow triangles) AOT timeseries, showing a large overestimation of the retrieved values.

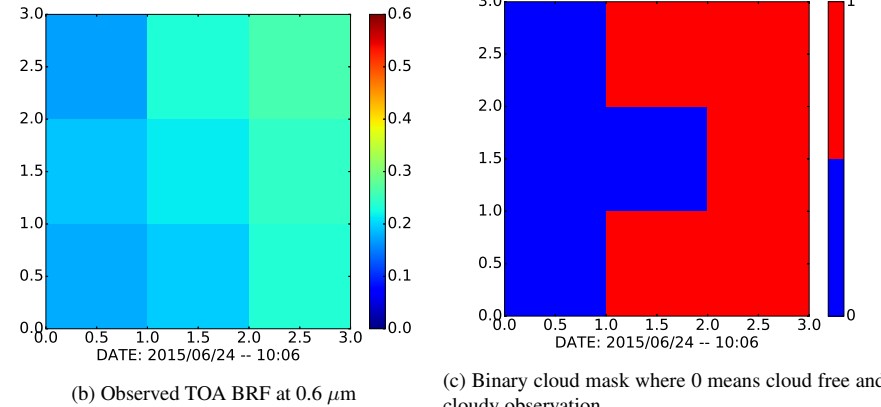

(b) Observed TOA BRF at 0.6 $\mu$m

(c) Binary cloud mask where 0 means cloud free and 1 cloudy observation.

Fig. 16: Example of possible cloud contamination from SEVIRI observations iver a 3x3 pixel window on the 24th of June over Burjassot, Spain. The upper panel show the CISAR retrieval (blue dots) and AERONET data (yellow triangles) over the central pixel. The bottom left and right panel show the TOA BRF and the cloud mask over the 3x3 pixel window respectively.


(coarse mode) events (Fig. 15). However, when the observation does not carry enough information, the CISAR algorithm relies on the prior information, this being an annual climatology value, hence normally lower than 0.2. Although the associated uncertainty is very large, the prior information still impacts the retrieved value. Some example of CISAR ability to detect high AOT are shown in the Supplement. The overestimation for low AOT might originate from undetected or neighbouring clouds. Cloud contamination can in fact be observed even few kilometres distant from clouds, increasing the AOT (Chand et al., 2012). Figure 16 shows an example of cloud contamination for the Burjassot AERONET station, Spain, for SEVIRI observations. Here the CISAR overestimation for low AOT values is clearly visible on June 24th. Figure 16 shows the TOA BRF and the cloud mask on a 3x3 pixels windows. The processed pixel is surrounded by cloudy pixels, probably leading to cloud contamination in the central pixel and consequently to the overestimation of the retrieval. Satellite measurements, being applied to pixels of few kilometers, are thus more likely to be affect by cloud contamination than ground observations. The probability to have pixels of that size contaminated by small undetected cloud could also explain this overestimation. CISAR potential to discriminate between the fine and coarse mode is analysed next. Figure 17 shows the fine and coarse mode ratio distribution related to AERONET (in red) and CISAR retrieval for SEVIRI (in blue) and PROBA-V (in green). It can been seen that the distribution related to CISAR retrievals from SEVIRI observations is in good agreement with the one associated to AERONET observations, whereas the retrieval from PROBA-V observations seems to underestimate the fine mode concentration for $\tau_F/\tau_C > 2$. The percentage of cases where CISAR succeeds in retrieving a predominant fine mode contribution to the total AOT, *i.e.*, $\tau_F/\tau_C > 1$), is in fact equal to 81% when the retrieval is performed

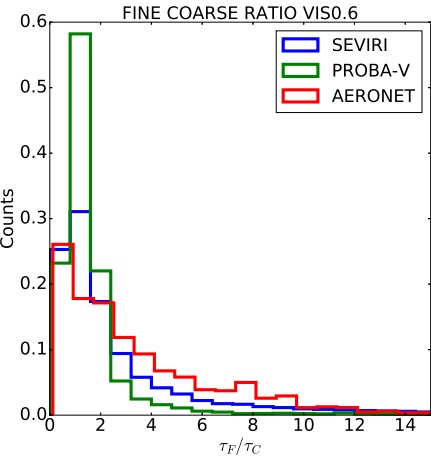

Fig. 17: Fine-coarse mode ratio distribution at 0.6 μm for AERONET (red), CISAR applied to SEVIRI (blue) and PROBA-V (green) observations.



on SEVIRI acquisition and 68% when CISAR is applied on PROBA-V data. The latter represents an improvement with respect to the Land Daily Aerosol (LDA) algorithm (Wagner et al. (2010), Table 4) where particles retrieved by AERONET as spherical were correctly characterise by the algorithm
only in the 12% of cases. This represents a decisive advantage of the proposed approach with a continuous variations of the aerosol properties in the solution space as opposed to the use of a limited number of aerosol classes as in Wagner et al. (2010). The coarse particle characterisation appears to be more challenging for both satellite. The percentage of cases where the coarse mode is correctly retrieved as predominant is 43% and 27% for the retrieval from SEVIRI and PROBA-V observations
respectively. The less accurate retrieval of the coarse mode compared to the fine mode is expected, as the considered wavelengths are less sensitive to the radii in the range of the coarse particles than to those of fine ones (Torres et al., 2017). A similar behaviour is also observed in Table 9 where the median magnitude of the coarse mode Jacobian is more than twice smaller than the fine mode Jacobian.

**6.2   Single scattering albedo and asymmetry factor**

In Section 3.2 the solution space defined by the aerosol classes vertices has been described. CISAR retrieves the averaged SSA and asymmetry factor within this solution space from Eq. 8 and 9 of Part I, *i.e.*, as linear combination of the micro-physical properties of each selected aerosol vertex. Figure 18 and 19 show the SSA and asymmetry factor distribution related to the AERONET data and
CISAR retrieval from both SEVIRI and PROBA-V observations. The three datasets show similar distributions, although spikes can be observed at the extremes of the CISAR retrievals distributions.

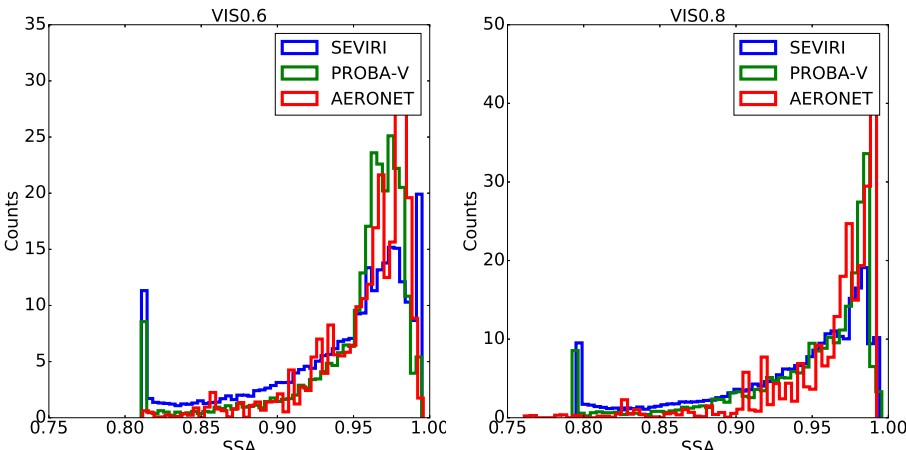

Fig. 18: SSA distributions at $0.6\mu$m (left panel) and $0.8\mu$m (right panel) for AERONET (red), CISAR applied to SEVIRI (blue) and to PROBA-V (green).





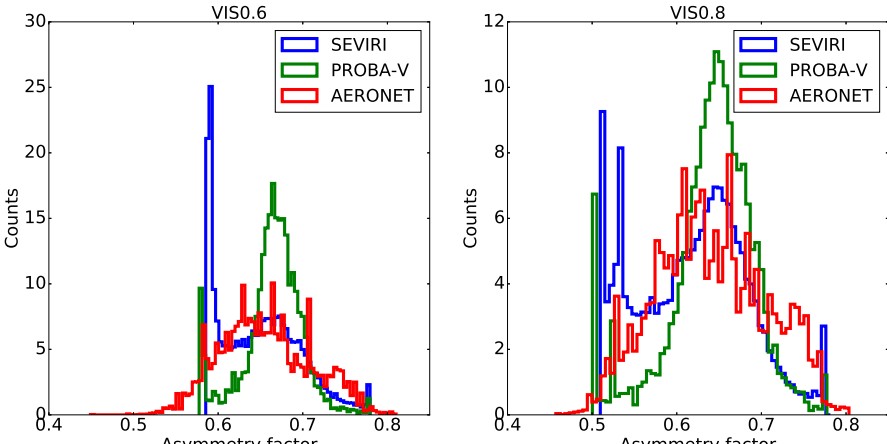

Fig. 19: Same as Figure 18 but for the asymmetry factor.

These spikes correspond to the sides of the triangle delineated by the three selected vertices shown in Fig. 4: when the actual AERONET solution is located outside the solution space, CISAR cannot reach it and the retrievals falls on the solution space boundaries, hence the spikes. The aerosol ver-

tices selection as in Fig. 4, *i.e.*, made to encompass about the 90% of the solutions as retrieved from AERONET inversion product, is conceived to limit the number of occurrences of these spikes. In Fig. 19 it can be noticed that the $g$ parameter distributions obtained from CISAR applied to PROBA-V observations is much narrower than the same distribution related to AERONET and CISAR applied to SEVIRI retrievals. This is in line with what has been seen in Section 6.1 on the poorer

CISAR performance in retrieving the predominant mode when applied to PROBA-V observations rather than SEVIRI ones. In fact, as in computing $g$ the aerosol size is the most important parameter to measure (Andrews et al., 2006), an inexact estimation of the dominant mode (fine or coarse mode) leads to an erroneous measurement of the asymmetry parameter.

**6.3   Surface bihemispherical reflectance**

CISAR BHR, computed from the RPV parameters, is compared with MODIS Land product (Schaaf and Wang, 2015). To account for the different spatial sampling, the MODIS data have been averaged on 5x5 km and 3x3 km for the comparison with the retrievals from SEVIRI and PROBA-V respectively. The results of this comparison are shown in Table 11 in terms of correlation, RMSE, and MAE. CISAR results show a high correlation with the MODIS product, higher than 0.800 in all the

processed spectral bands, except the PROBA-V NIR band, which shows a correlation equal to 0.774. The density plots of CISAR BHR retrieval against MODIS data are included in the Supplement for all processed bands, for both satellites. Despite the instruments differences discussed in Section 2.5,



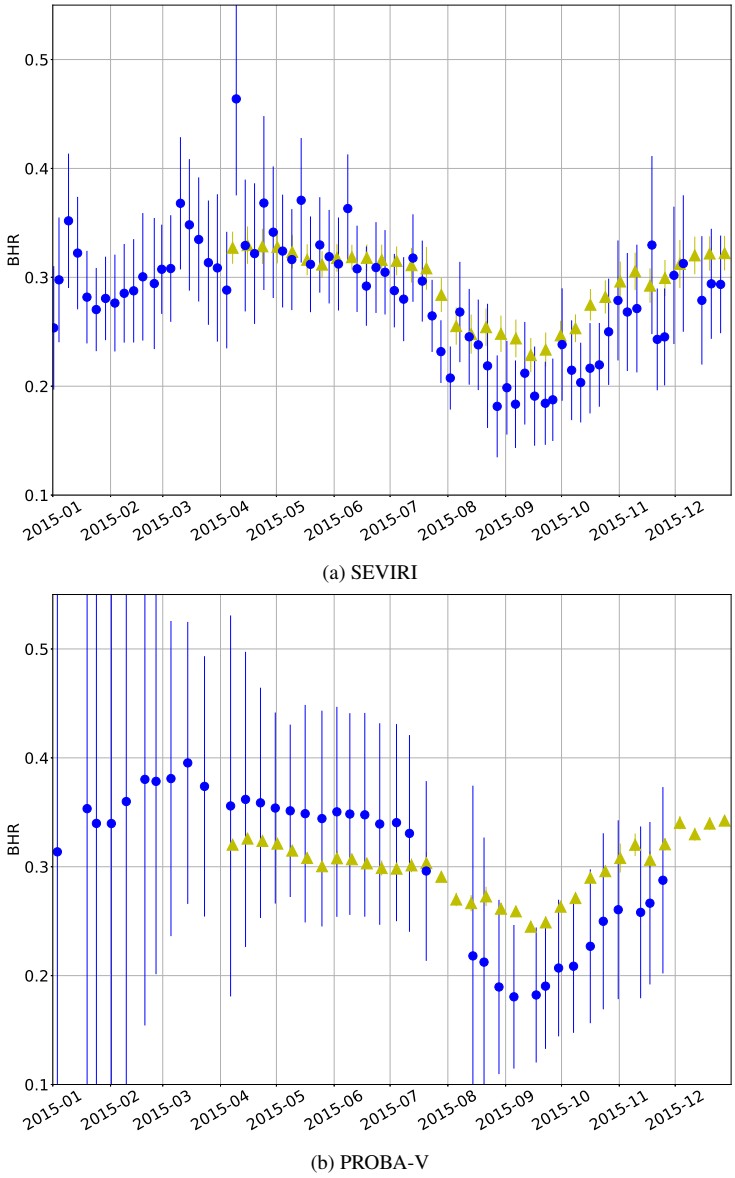

(a) SEVIRI

(b) PROBA-V

Fig. 20: CISAR retrieved BHR (blue dots) from SEVIRI (upper panel) and PROBA-V (lower panel) and MODIS Land Product (yellow triangle) averaged on SEVIRI and PROBA-V pixels over Zinder Airport (Niger, Africa). The results are shown for the sensors band centred at 0.6 $\mu m$ over 2015. The vertical bars represent the CISAR retrieval uncertainty.

CISAR retrievals and the MODIS Land Product dataset show similar seasonal trends. Figure 20 shows the BHR timeseries over Zinder Airport (Niger, Africa), as retrieved from the CISAR algo-




rithm applied to SEVIRI and PROBA-V observations and from MODIS Land Product. The rainy
season, going from May 20 to October 5 (Weatherspark.com, 2018), is distinguishable in all the
datasets, although CISAR retrieves a larger seasonal variation with respect to MODIS product. The
effect of the updating mechanism on the surface prior described in Section 2.4 is also visible, espe-
cially in Fig. 20b, where the retrieval uncertainty decreases in time, as the prior information on the
surface is better defined.

## 7   Discussion and conclusion

This paper describes and evaluates the CISAR algorithm when applied to satellite observations ac-
quired from geostationary and polar orbiting instruments. The theoretical aspects of CISAR, a new
generic algorithm for the joint retrieval of surface reflectance and aerosol properties, with continuous
variation of all the state variables in the solution space, are described in Part I. In the latter CISAR is
applied to simulated noise free observations in the principal plane. This paper provides an evaluation
of the algorithm in non ideal situations, *i.e.*, when applied to actual satellite observations acquired
from both geostationary and polar orbiting satellites, namely SEVIRI and PROBA-V.

The proposed retrieval method relies on an OE approach which consists in the inversion of FAS-
TRE, a simple radiative transfer model composed of two horizontal layers. The first step of CISAR
algorithm evaluation consists thus in the evaluation of the inverted forward model (Section 2.5). The
FASTRE model is accurate within 1% to 3% percent when compared to a complex 1D radiative
transfer model. Higher uncertainties are observed in spectral bands affected by water vapour as a
result of the limited vertical discretisation. The FASTRE model capabilities to simulate actual ob-
servations has also been evaluated but revealed relative bias larger than 5%. This poor performance
partially results from the lack of accurate description of the state variables at the moment of the
satellite overpasses.

The analysis of the information content of the satellite observations is performed in Section 4.
Despite the PROBA-V instrument has one blue channel which is not present on SEVIRI, the frequent
revisit rate of the latter provides more information for the retrieval of surface reflectance and aerosol

Table 11: CISAR retrieved BHR from actual observations comparison with MODIS in all the pro-
cessed bands.

|  | SEVIRI | | | PROBA-V | | | |
|---|---|---|---|---|---|---|---|
|  | 0.6 $\mu$m | 0.8 $\mu$m | 1.6 $\mu$m | 0.6 $\mu$m | 0.4 $\mu$m | 0.8 $\mu$m | 1.6 $\mu$m |
| Number of points |  | 7409 |  |  | 744 | | |
| Correlation | 0.925 | 0.820 | 0.860 | 0.763 | 0.891 | 0.774 | 0.890 |
| Root Mean Square Error | 0.045 | 0.067 | 0.080 | 0.030 | 0.051 | 0.092 | 0.085 |
| Mean Absolute Bias | 0.038 | 0.053 | 0.067 | 0.026 | 0.042 | 0.064 | 0.068 |





properties than the former instrument.

CISAR retrieval is finally evaluated against independent datasets. The retrieved AOT is compared against AERONET data. A specific QI has been developed to disregard suspicious retrieval and has been applied in this analysis. With a RMSE of 0.173 for SEVIRI and 0.214 for PROBA-V CISAR

shows better performances when applied on geostationary data, as expected from the analysis in Section 4. CISAR retrieves the micro-physical aerosol properties assuming a linear behaviour of $g$ and $\omega_0$ in the solution space in Fig. 4; although this assumption is not exactly true when far from pure single mode situations, CISAR retrieved aerosol properties distributions are in good agreement with the AERONET inversion products, especially when the algorithm is applied on geostationary

observation, as discussed in Section 6.2. These differences are explained by the different information content associated to the satellite observations acquired with different orbits. For both satellites, CISAR discrimination between fine and coarse mode is improved with respect to the LDA algorithm (Wagner et al., 2010), as the continuous variation of the aerosol properties in the solution space allows a more accurate retrieval of the micro-physical properties with respect to LUT-based

approaches. The CISAR surface albedo is compared with MODIS product, showing a correlation higher than 0.77 in all processed bands.

Several aspects of the new CISAR algorithm would still require additional efforts to improve its performance. The capability of the FASTRE model to provide an unbiased estimation of observed TOA BRF still remains to be clearly demonstrated. Cloud effects on the AOT overestimation at low

optical thickness would also deserve additional work. The analysis of the Jacobian median values has revealed the very small magnitude of the fine and coarse mode AOT. Spectral and temporal constraints of the AOT variability play therefore a critical role in supporting the assessment of aerosol properties. However, these constraints might lead to an underestimation of AOT retrieval at large values.

As pointed out in Part I, the limited number of state variables retrieved by CISAR allows the same algorithm to be applied on sensors which have not been originally designed for aerosol or surface albedo retrieval. The possibility to apply the same algorithm on data acquired by different sensors for the retrieval of several ECVs presents a decisive advantage. It provides radiatively consistent ECVs between themselves derived from different sensors. Conversely, the use of separate methods for the

retrieval of different variables might lead to a radiance bias, which has to be corrected preliminary to the assimilation of these variables (Thépaut, 2003). The effort for the assimilation of surface and atmospheric products could be reduced if the different ECVs are consistently derived with one single algorithm. The consistent retrieval of the state variables and the algorithm applicability to different sensors represent an important advantage for the Numerical Weather Prediction (NWP) community,

whose main future challenges are related to a more consistent retrieval of Earth's system components and to the availability of more satellite data. The latter makes desirable an algorithm applicable to different types of orbits and bands in order to have continuity in the dataset.



## 8 Supplement

Includes the scatterplots between the BHR retrieved by CISAR and the one delivered in the MODIS
product (Fig. S1, S2) and few examples of CISAR high AOT retrievals in comparison with
AERONET data.

## 9 Acknowledgements

*Acknowledgements.* This work has been performed in the framework of ESA projects aerosol_cci2 and PV-LAC
under the contract 4000109874/14/I-NB and 4000114981/15/I-LG respectively.



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
