# Peer review of "Joint retrieval of surface reflectance and aerosol properties with continuous variation of the state variables in the solution space: Part 2: Application to geostationary and polar-orbiting satellite observations"

_Atmospheric Measurement Techniques, 2018_

## Referee Comment (RC1) · Anonymous Referee #1 · 21 Aug 2018

As this manuscript is a follow-up to Part I, I have proceeded to a full review which will be posted in the next step.

---

## Referee Comment (RC2) · Anonymous Referee #1 · 21 Aug 2018

I think Part II does a rather nice job in introducing the different sources of uncertainties for SEVIRI and PROBA-V (uncertainty assessment - a necessity - is too often neglected). As a general impression though, the writing seems more of a lab "living" log, i.e. notes accumulated as the work was being carried out. You'll notice how many of my highlighted comments aim at condensing the text as the reader might get lost in

details that are often redundantly expressed. Please try and be concise. Every time you start a sentence with "In other words,", ask yourself why you need to re-explain what you just said. As a reader, I had the impression of re-living the struggles to make sense of results, and learning a lot about the things that can go wrong while developing an algorithm, rather than walking confidently away with a message on original and reliable results. This is also reflected in grammatical hurdles. The manuscript should be proofread before submission; see countless instances of 1) "on" instead of "in"; 2) excessive use of "i.e.", "the former" or "the latter", "ones", "it can be seen", "it should be noted", or references to other sections when not really needed; 3) missing plurals; 4) missing articles; 5) "Section" and "Figure" instead of "Sec." and Fig." according to the journal's guidelines; 5) the term "miss-fit".

A pdf with detailed comments is attached. In the following are some other general comments:

Line 116: BRF needs be defined.

Section 2 concludes with "More effort would be needed to demonstrate that the forward RTM is unbiased". This is the kind of sentences disseminated all over Part I that shake confidence in the method. This particular sentence alone gives the impression that the whole method is systematically flawed. Unless the bias is quantified being negligibly small what should the reader take away from this message? As remarked above, the draft goes at quite a length in explaining different sources of uncertainties smaller than 1%; if this last bias is larger, it would cast quite a different light on the accuracy of the method.

Figure 4. This way to depict the subspace of solution is misleading. For example, the way you have things set up now, the magenta triangle does not include the peak of the distribution, with omega>0.98 and g~0.75. Lots of aerosol types are found in this region. How do you deal with this?

Regarding the graphs: - enlarge ALL font sizes. Things are barely legible. - please

[Figure]

take away the grid lines form all graphs. They're extremely confusing and provide no useful information. - many figures feature symbols in yellow green and violet, which are among the least differentiable colors (ask color-blinded folks). Try with more opposing colors like blue (or black) and red.

Fig 6. : merge the two panels into one, since you compare Carpentras with Zinder.

Line 280-281: this statement is simply not true and has to be reversed. While it is true that the diffraction peak is very sensitive to size, the backscattering contains tons of information (pretty much everything else). We wouldn't be doing space-based remote sensing otherwise!

Line 283-284: what was the retrieved optical depth for this day? AND AT WHAT WAVE-LENGTH? This is an essential piece of information. How would the figure change if the AOT is 0.05 or 0.8? A discussion on the linearity of the AOT Jacobians is due in the text.

The "Principle" in Sec. 5.1 needs to be explained better. Please re-elaborate lines 320-330. I simply couldn't get why the number of could-free pixels should be proportional to the quadratic sum of the mismatch between simulation and observation. Even in the rest of the section, I lost the logical thread. The QI/p tests part is very mysterious, I just did not get it. "QI" is not even defined, and there's no explanation of its range of values. Please review the whole text and try to make it more understandable. Also, "miss-fit" is not a correct terminology; change to "mismatch" or something else. Little to no guidance is offered for the comprehension of Fig. 11. WHEN IS A RETRIEVAL DEEMED SUCCESFULL?

In both manuscripts, it's never clear if CISAR can be applied to water and land indifferently. This should be made more clear throughout.

The approximation of a two-layer atmosphere is not discussed. In fact, it could be a reason for the algorithm failure in many cases.

Overlap graphs in Figs. 9 and 10 so as to make one figure only

Discussion following Eq. 10: it has to made clear if you're talking about "entropy" or "entropy difference" between pre and post retrieval.

Sec. 5.2 is "Theoretical Concept" and comes after Sec. 5.1, i.e., "Principle". I see no point in fragmenting the text this way. Please condense the sections.

Line 360: it remains a mystery why a cloud mask is not applied.

Line 456-459. This is one of my most important comments. After the manuscript goes to a great length in describing a very elaborate way to aid the retrievals with "tests", the results presented in Fig. 14 are clearly not satisfactory (a look at the correlation coefficients immediately tells that the algorithm is not retrieving appropriate AOTs). Then it is commented that at high AOTs the algorithm might fail (then why all the tests?), but that's not too worrisome since it is better if it performs accurately at low optical depths, which are more typical. I might agree with that, but then I have to ask 1) how do you deal with the fact that the 1:1 correlation is as poor at low optical depths; and 2) why the only AOT used for testing was 0.4 in part 1.

Line 477-480. I don't understand these comments about Fig. 17. CISAR/SEVIRI is in very good agreement? As CISAR/PROBA-V, it misses the peak of the distribution. Also, CISAR/PROBA-V is said to be underestimating the fraction but so does CISAR/SEVIRI. The significance of the ratio should also be discussed. What are typical ranges?

The relative magnitude of those"spikes" in Figs. 19 and 20 are worrisome. For the causes you attribute, shouldn't they confirm that your choice of the three vertices is inadequate?

Line 487: I take the chance here to expand on previous comments. "Coarse mode characterization" is very far-fetched. The algorithm is not so much retrieving surface and aerosol properties, as much as two aerosol radiative properties and a set of RPV
Interactive
comment

parameters white variability has not been ascertained. Even here, you've already got problems with unreliable retrievals of fine-to-coarse ratio, so much that you focus on the ratio being less or larger than 1. For these reasons, the title sounds a bit pretentious and should be adjusted accordingly. Omega and g are properties but based on the current title nowadays most readers would expect an extended set of microphysical and optical properties.

Sec. 6.3: how about Carpentras?

Line 545-547: This is either too obvious or a concept I don't get. You don't describe state variables, you retrieve them, so isn't just that the algorithm fails?

The manuscript should report complete statistics on the number of analyzed scenes, so that the retrievals can be put in context. I'm not sure this is what happens in Table 11.

Please also note the supplement to this comment:
https://www.atmos-meas-tech-discuss.net/amt-2018-265/amt-2018-265-RC2-supplement.pdf

**Supplement:**

This is just a preview and not the published paper.

[revised manuscript text omitted]

---

## Referee Comment (RC3) · Anonymous Referee #2 · 30 Aug 2018

This paper evaluates a retrieval of surface reflectance and aerosol optical properties proposed in its accompanying paper. A quality control system is outlined and the sensitivity of the retrieval discussed. Retrievals from SEVIRI and PROBA-V data during 2015 are compared to AERONET, showing a disappointing AOD retrieval, and the MODIS Land Product, which are rather more satisfying.

[Figure]

I wish to clearly state that I quite like the idea behind this algorithm. Single-scattering albedo and the asymmetry parameter provide a theoretically superior state space in which to evaluate aerosol retrievals and I would love to see (and do) more research around this idea. I am always pleased to see a discussion of information theory in an atmospheric science paper and more validation papers should discuss uncertainty. I want to see this pair of papers eventually published.

My issue is that I see no evidence that this algorithm currently produces acceptable results. Fig. 14 is not good. It's not bad enough to imply your technique is without merit, but if that's the only plot you're going to provide, you will struggle to attract interest in this algorithm as your correlation, bias and RMSE are worse than most products I've encountered. At the very least, you need to find some circumstances where your retrieval's ability to mix aerosol types produces a better retrieval than a more developed product (e.g. MODIS collection 6.1 or the Swansea University product from Aerosol CCI). Maybe biomass burning emissions from Africa or the industrial regions of China?

Also, the heritage of the algorithm and the plots in the supplement imply this method is a much better retrieval of the surface than of aerosol. I would warm to the paper more if it was arguing that you made a slightly better aerosol retrieval without harming the surface product rather than the current structure, which implies you were trying to make an aerosol retrieval and skims over the significant limitations in your current results.

A list of my more major concerns follows, attempting to only repeat points I made in my reviews of Part 1 and the comments of Reviewer #1 for emphasis.

§4 Though I'm pleased to see a discussion of information content in an atmospheric science paper, yours is rather unusual. You're using the magnitude of the Jacobian to argue which terms are the most important. However, the Jacobian has units and so the magnitude of different terms isn't direct comparable.

To illustrate, consider Fig. 5, which you use to argue that $\rho_0$ is a more dominant driver of changes in TOA radiance than $\theta$. A small change in surface reflectance

could be of order $10^{-3}$, which would produce a change of about $10^{-3}$ in $\mathbf{y}$ (as the Jacobian is approximately unity). A small change in viewing angle could be $1°$ and, if the Jacobian shown was in units of degrees, that would imply a change of $-0.2$ in $\mathbf{y}$, which is much larger than that for $\rho_0$. (The change is still larger if the units are radians.) The value of the Jacobian must be scaled by an appropriately small change to be compared to other values.

Optimal estimation already has a mechanism to evaluate this. It's called the averaging kernel and Eq. 2.78 of Rodgers (2000) defines it as,

$$\mathbf{A} = \left(\mathbf{K}^T\mathbf{S}_\epsilon^{-1}\mathbf{K} + \mathbf{S}_a^{-1}\right)^{-1}\mathbf{K}^T\mathbf{S}_\epsilon^{-1}\mathbf{K}.$$

You likely already calculate this when determining the entropy (see Eq. 2.80). A row of the averaging kernel summarises the contributions of each state vector element to the retrieval of each other variable while the diagonal elements quantify the reliance on the prior. (Things are slightly complicated by the addition of smoothing, $\mathbf{H}$, terms to your cost function. The difference is subtle; ask Oleg Dubovik about it.) For your retrieval, I would expect the diagonal of $\mathbf{A}$ for $\rho_0$ to be close to one and $k$ to be closer to zero. It would also illustrate the interdependence of the different terms.

I don't know if the average reader would find such an analysis easier to understand. Averaging kernels, though very powerful, are confusing. I tend to put them in supplementary material for people that care to find.

If you don't switch to averaging kernels, label your plot axes as derivatives rather than Jacobians (e.g. the $x$-axes on Fig. 6 is $\frac{dy}{d\tau}$) so readers have some chance of understanding what's being plotted.

More practically, I'd say a superior test to use in §5.2.4 would be the number of degrees of freedom for noise (e.g. $n - \operatorname{tr}\mathbf{A}$).

§4 More generally, I'm not sure why this section is so long. It's worthwhile to point out that the retrieval's sensitivity is a function of what is observed, but there must

be a more efficient way to show that the retrieval has minimal sensitivity at some times of day/year.

Tab.2 This is a substantial problem. You should be more upfront about the current limitations of your method and outline in more detail what you intend to do about them. There's nothing wrong with incremental progress. This also affects L568.

Fig.4 I agree with the other reviewer in wondering why you selected vertices that exclude a significant population of observed aerosols.

L280 I partially disagree with the other reviewer. For aerosol, there is less information content in the backscattering direction. This is why the orientation of the second view was flipped from AATSR to SLSTR. The instrument now views backscattering in the Northern Hemisphere, reducing the influence of aerosol on the signal and improving the quality of surface products in the region of the world that contains most of the humans.

L299 I strongly suspect that there is less information content in the polar data because you ascribed more uncertainty to it ($\sigma_c$ and $\sigma_\theta$), not because of anything intrinsically advantageous to the geostationary view. This affects your conclusions on L555 and L561. (My opinion is that geostationary data is superior when you need temporal resolution and polar data superior when you need global coverage.)

L321 Do you mean that the magnitude of the cost increases with the number of observations because there are, well, more observations?

– L297 of Part 1 addressed something similar by putting a scaling into the cost function; you could do that.

– The cost function is (theoretically) a $\chi^2$ distribution with a number of degrees of freedom equal to the number of observations. Using that model, the cost can be converted into a probability that the fit is coincidental and a

threshold for retrieval quality defined in terms of that (for example, keeping only retrievals with less than a 5 % probability of being the result of chance).

– Regardless, I agree that filtering by cost alone will not identify retrievals with minimal sensitivity.

§5.2 This section is very difficult to follow and needs redrafting with help from someone unfamiliar with the method. Switching between $p$, $q$, and $QI$ doesn't help, especially when 1 is a good value for one while 0 is a good value for the other. It would be substantially easier to follow if you provided a decision tree.

§5.2.3 Though I understand the motivation behind this test, I should point out that $\frac{y_m - y_0}{\sigma_0}$ is normally distributed. As such, 31.8 % of observations would be expected to fail your test by simple chance.

L360 I agree with the other reviewer that the lack of discussion of a cloud masking is surprising. PROBA-V lacks thermal channels, making it difficult, but you have no problems on SEVIRI.

L425 This extra test should have been mentioned back in §5.2.5. More justification of this work around is necessary.

L453 A factor of two is not a 'slight' overestimation and the fact that your retrieval was this bad eight years ago does not forgive it's failure now.

L478 That isn't good agreement. A good agreement can be seen between the red and green lines in Fig. 18(a).

§6.2 These comparison look good! Why not give us a version of Fig. 14 for SSA and $g$? Considering they're what you retrieve, I wouldn't be surprised if you could estimate them better than you could AOT. Wouldn't make me think the product was any better as most users want AOT, but they aren't many global SSA datasets

and if you could provide one, even if it's very uncertain, that would be something worth writing about.

Some more minor comments,

L116 There are many potential calibration methods for SEVIRI. If you're using IMPF or GSICS, could that be mentioned explicitly? If you're using something in-house, a citation would be appreciated.

L145 Why make this approximation? Is the calculation of the other terms computationally expensive?

Eq.6 This seems a strange choice. Why not the standard deviation or interquartile range or a constant value based on climatology?

§2.4(1) What's the value of $N_{min}$? Why increase the uncertainty by 5 % per day rather than any other amount?

§2.5 I'd actually prefer to see a thorough sensitivity study of bias as a function of the various parameters rather than the simple 1 - 3 % uncertainty you've added, but that can be in a third paper.

P12L2 In my experience, the first guess is set to reduce the number of iterations needed to reach a solution. Avoiding local minima involves checking the shape of state space around the final solution or annealing (i.e. running multiple retrievals on the same data).

Eq.8 So you're using a different first guess for even and odd numbered time steps? That's peculiar and, on its own, I don't see how it avoids local minima.

§4 The third paragraph covers four pages. Perhaps split it up.
P17L1 As the sensitivity drops through the day, I would expect the uncertainty to increase.

L351 What is the maximum number of iterations?

L352 Could you clearly state that $p_0 = 1$ in all other circumstances. I wasn't certain of that till I got to Eq. 15.

§5.2.2 Did you ever explore using the a priori cost for this test (i.e. the difference between the retrieval and the prior)?

Eq.11 Aren't the $y$ terms vectors? If so, wouldn't this require some sort of sum?

L371 Didn't you have to calculate the full Jacobian to perform your inversion? I see your point, but this is a lot of explanation for why you don't use something you should already have.

L379 I assume that if I ask for a justification of this statement, I will be told to go look at your papers from 2010 so I will make this sarcastic remark instead.

Eq.15 For the sake of future readers' comprehension, please restrict $q_i$ to the range $[0, 1]$ and make $QI$ a simple product rather than use the difficult to comprehend $1 - \max(q, 1)$ construction.

L409 Please specify this sigmoid function (or at least give it's width).

Fig.14 Can we please have a version of this plot as a 2D histogram in the supplement, similar to the ones already there for the BRF?

- The $y$-axis of Figs. 5, 6, 9, 10, 15, 17, 18, 19 should probably be 'Fractional counts' considering they clearly have non-integer steps.

Fig.16 (b) and (c) aren't that interesting or helpful. Perhaps make (c) an inset in (a).

The English is easily understood but I agree with Reviewer #1 that it reads more like speech than text. There are too many interrupting statements and initial qualifiers. As the other reviewer provided extensive technical corrections, I list here only those issues that I noticed in the course of reading.

L1 Move 'simultaneous retrieval' to before 'Aerosol'.

L4 demonstrate CISAR's applicability (Similar corrections on L24, 464, 474, 583.) This correction isn't universally preferred and an alternative would be 'demonstrate the applicability of CISAR applicability to'.

L32 with a revisit time in the range of several tenth tens of minutes

L33 limited field of regard view (similar corrections in Fig. 1 and on L67).

L33 that many different instruments with different sub satellite locations are needed

L34 Earth. The poles cannot

L66 applied on to observations

L81 in Eq. 17 in of Part 1

L82 composed by of the radiometric

L87 observation in the near-real time

L90 with a 15 minutes minute repeat

L137 The reference to Dee et al. lacks a year; it's 2011.

L142 strongly affect affects the CISAR

L152 is not supposed expected to undergo

L170 FASTRE  must know

Fig.3 during  5 days

L210 For  polar

P12L7 computational  performance, the

L245 latter consists  of the minimisation

L253 reflectance shape  is more

L261 When the magnitude of AOT

L305 smaller  values than

L307 polar orbiting  ones, the

L340 to  define a $QI$

L345 been defined  through an analysis

L365 For the inversion  to be successful

L370 Performing a test  on the Jacobians

L373 state variables, for which

L382 applied  to the AOT

L391 parameters,  computed as follows

L415 been chosen by observing

L456 the algorithm  from fitting rapidly

L458  of AOT  is normally smaller

L464  Some  examples of

L465  overestimation  of low AOT

L466  observed even  few kilometres  from

L472  measurement,  observing areas of a few kilometres . . . to be  affected by

L484  correctly  characterised by

L488  for both  satellites. The

L498  linear  combinations of

L521  plots of the CISAR BHR

L539  consists  of the

L540  step of the CISAR

L541  evaluation consists  of the

L549   Though the PROVA-V

L552  The CISAR retrieval is  evaluated

L553  suspicious  retrievals and

L559  is applied  to geostationary  observations

L565  with the MODIS

L576 applied  to sensors

L577 algorithm  to data

---

## Author Comment (AC1) · 5 Oct 2018

1. I think Part II does a rather nice job in introducing the different sources of uncertainties for SEVIRI and PROBA-V (uncertainty assessment - a necessity - is too often neglected). As a general impression though, the writing seems more of a lab "living"

log, i.e. notes accumulated as the work was being carried out. You'll notice how many of my highlighted comments aim at condensing the text as the reader might get lost in details that are often redundantly expressed. Please try and be concise. Every time you start a sentence with "In other words,", ask yourself why you need to re-explain what you just said. As a reader, I had the impression of re-living the struggles to make sense of results, and learning a lot about the things that can go wrong while developing an algorithm, rather than walking confidently away with a message on original and reliable results. This is also reflected in grammatical hurdles. The manuscript should be proofread before submission; see countless instances of 1) "on" instead of "in"; 2) excessive use of "i.e.", "the former" or "the latter", "ones", "it can be seen", "it should be noted", or references to other sections when not really needed; 3) missing plurals; 4) missing articles; 5) "Section" and "Figure" instead of "Sec." and Fig." according to the journal's guidelines; 5) the term "miss-fit".

The paper has been improved thanks to the detailed comments in the annotated PDF which have been implemented in this revised manuscript. Our replies are directly included in that document. Occurrences of "i.e.", "the former", "the latter", ..., has been drastically reduced. The abbreviations are now in agreement with the house standards. The term miss-fit has been replaced by mismatch. The grammatical and styling suggestions have been implemented.

2. Line 116: BRF needs be defined.

TOA BRF are now spelled out separately at line 115 (now 113)

3. Section 2 concludes with "More effort would be needed to demonstrate that the forward RTM is unbiased". This is the kind of sentences disseminated all over Part I that shake confidence in the method. This particular sentence alone gives the impression that the whole method is systematically flawed. Unless the bias is quantified being negligibly small what should the reader take away from this message? As remarked above, the draft goes at quite a length in explaining different sources of uncertainties

smaller than 1%; if this last bias is larger, it would cast quite a different light on the accuracy of the method.

The accuracy of the FASTRE model has been demonstrated only against comparisons with a reference 1D RTM. We underestimated the efforts needed to demonstrate that this model can fit actual satellite data. It would require a detailed characterization of the surface and atmosphere at the overpass time which is currently lacking. It is beyond the resources we had to perform these studies and would probably require a paper of its own. We have therefore decided to remove this paragraph for the time being. We have not found in the literature similar attempts. The assessment of FASTRE uncertainty is now described as in Part I, comparing FASTRE simulations with a reference RTM (RTMOM), and the limitations due to the 2-layer approximation are discussed. Lines 45-46, 500-501, 545-547 and 569-570 have also been removed.

4. Figure 4. This way to depict the subspace of solution is misleading. For example, the way you have things set up now, the magenta triangle does not include the peak of the distribution, with omega>0.98 and gâĹij0.75. Lots of aerosol types are found in this region. How do you deal with this?

The aerosol vertices have been adjusted to include the peak of the distribution. Line 206-207 of the revised paper now read "The selected CISAR vertices defining the solution space cover about the 80% of possible solutions (black triangle)."

5. Fig 6. : merge the two panels into one, since you compare Carpentras with Zinder.

The merged figure looks quite confusing (Fig. 1). Even changing colors and line styles it would still not be very easy to understand. Keeping the separate panels appears a better choice.

6. Line 280-281: this statement is simply not true and has to be reversed. While it is true that the diffraction peak is very sensitive to size, the backscattering contains tons of information (pretty much everything else). We wouldn't be doing space-based

remote sensing otherwise!

Please see answer reviewer 2 regarding line 280.

7. Line 283-284: what was the retrieved optical depth for this day? AND AT WHAT WAVELENGTH? This is an essential piece of information. How would the figure change if the AOT is 0.05 or 0.8? A discussion on the linearity of the AOT Jacobians is due in the text.

The retrieved optical thickness at 0.55 $\mu$m is now shown in the plot (Fig. 8) at each observation. A more detailed discussion on the AOT Jacobians magnitude can be found in Luffarelli et al. 2016 (the reference has been added in the text).

8. The "Principle" in Sec. 5.1 needs to be explained better. Please re-elaborate lines 320- 330. I simply couldn't get why the number of could-free pixels should be proportional to the quadratic sum of the mismatch between simulation and observation. Even in the rest of the section, I lost the logical thread. The QI/p tests part is very mysterious, I just did not get it. "QI" is not even defined, and there's no explanation of its range of values. Please review the whole text and try to make it more understandable. Also, "miss-fit" is not a correct terminology; change to "mismatch" or something else. Little to no guidance is offered for the comprehension of Fig. 11. WHEN IS A RETRIEVAL DEEMED SUCCESFULL?

The whole Sect. 5 has been rewritten and it is now organized as follows: 5.1 Review of existing methods 5.2 Overview 5.3 Quality indicator tests 5.4 Quality indicator computation Section 5.3 now includes most of what was described in Sect. 5.4 which is now much shorter and, hopefully, readable. QI was defined in the introduction and according to the house standard does not have to be repeated. The term miss-fit has been replaced by mismatch. The QI/p tests part has been simplified, removing the qi definitions. Lines 346-348 (now 304-306) commenting Figure 11 (now Fig. 10) read now: "Figure 10 shows an example of the evaluation of the retrieved AOT against AERONET data for the mismatch test (3). As the mismatch increases, the correlation decreases,

while the RMSE shows opposite behaviour."

9. In both manuscripts, it's never clear if CISAR can be applied to water and land indifferently. This should be made more clear throughout.

Line 66 (previously 69) now reads "These targets span different geometries and land cover types (vegetation, urban, bare areas, water, mixed)". Table 1 includes both water and land cover type (it was already the case). Part I (lines 209-210) states that surface reflectance simulations over water are performed with the Cox-Munk model. However, in that case, surface reflectance is not retrieved but calculated on the basis of the surface wind considered as a model parameter.

10. The approximation of a two-layer atmosphere is not discussed. In fact, it could be a reason for the algorithm failure in many cases.

The two-layer approximation has been inherited from the approach proposed by Pinty et al. (2000) and Govaerts et al. (2010). Section 2.5 now discusses the limitations of the two-layer approximation. Lines 162-170 of the revised paper read: "The forward model uncertainty is lower than 3% in all processed bands, presenting its largest value in the SEVIRI VIS0.8 band, the most affected by water vapour absorption (Table 4). The FASTRE two-layer approximation of the atmosphere does not allow a correct discretisation of the water vapour vertical profile and, thus, a correct characterisation of its interaction with the scattering particles. Moreover, the two-layer approximation assumes that the scattering particles are only present in the lower layer. Given the spectral behaviour of the AOT, this assumption leads to a higher uncertainty at wavelengths shorter than 0.4 $\mu$m (Seidel et al., 2010). Despite the limitations associated to the two-layer approximation, FASTRE uncertainty is in the range of 1% - 3% (Table 6), which is smaller or equal to the instrument radiometric noise."

11. Overlap graphs in Figs. 9 and 10 so as to make one figure only

Fig. 9 and 10 are merged in one figure with 2 panels showing, for both satellites, the

entropy related to the AOT and to the RPV parameters respectively.

12. Discussion following Eq. 10: it has to made clear if you're talking about "entropy" or "entropy difference" between pre and post retrieval.

The concept of entropy difference is never mentioned nor used in the paper. What it is used is the entropy, computed as in Eq. 10 after Rodgers (2000). The entropy is mathematically defined as the logarithmic ratio between the prior uncertainty and the posterior uncertainty. It thus measures the uncertainty reduction from the prior to the posterior.

13. Sec. 5.2 is "Theoretical Concept" and comes after Sec. 5.1, i.e., "Principle". I see no point in fragmenting the text this way. Please condense the sections.

Please see anwer to comment #8

14. Line 360: it remains a mystery why a cloud mask is not applied.

"Cloud contamination" has been changed to "cloud mask omission errors". A cloud mask is indeed applied, but omission errors might be present as discussed in Sect. 6.2 (previously 6.1).

15. Line 456-459. This is one of my most important comments. After the manuscript goes to a great length in describing a very elaborate way to aid the retrievals with "tests", the results presented in Fig. 14 are clearly not satisfactory (a look at the correlation coefficients immediately tells that the algorithm is not retrieving appropriate AOTs). Then it is commented that at high AOTs the algorithm might fail (then why all the tests?), but that's not too worrisome since it is better if it performs accurately at low optical depths, which are more typical. I might agree with that, but then I have to ask 1) how do you deal with the fact that the 1:1 correlation is as poor at low optical depths; and 2) why the only AOT used for testing was 0.4 in part 1.

The bias between the CISAR retrieval and the AERONET data is shown in Fig. 2, which shows different performances for SEVIRI and PROBA-V. These differences show that

the bias does not only depend on the CISAR algorithm itself, but also on the quality of the processed data. The green histogram shows the AERONET AOT distribution for each bin associated with the CISAR applied SEVIRI AOT product. It can be seen there only few points correspond to AOT>0.8 (less than 5% of the total number of observations), affecting the reliability of the statistics for high values of AOT. The histograms have been added in Fig. 14.

The CISAR AOT product shows overestimation at low AOT and underestimation at large AOT values. The overestimation rapidly decreases as the AOT approaches values of about 0.2. The retrieval is within the GCOS requirement (dashed lines) for 0.2 <AOT< 0.75. For SEVIRI, two factors might explain the overestimation of the retrieved AOT below 0.2. Firstly, most of the selected AERONET stations are located in Europe as can be seen on Fig. 1 of the revised paper, where the SEVIRI pixel resolution is about 5 x 8 km (as opposed to 3x3km at the subsatellite point) which is compared to AERONET point measurement. The probability of residual cloud contamination at this scale might thus explain part of the overestimation (Henderson and Chylek 2005, https://ieeexplore.ieee.org/document/1499014/authors#authors ). Secondly (and most likely explanation), it should be reminded here that SEVIRI shortest spectral band is 0.67. At low optical thickness, e.g., 0.1, the sensitivity to aerosol at 0.67 is about 2 times smaller than in the blue spectral regions and 1.5 smaller than in the red. A preliminary analysis revealed that the sensitivity of the TOA BRF to an increase of the AOT from 0.05 to 0.15 is responsible over dark surface to a change comparable with the magnitude of the radiometric uncertainty in the 0.67 $\mu$m band. Consequently, the retrieval in these cases essentially relies on the prior information despite the very large associated uncertainty (1.0 for the fine mode, 2.0 for the coarse mode). The prior AOT magnitude is taken from the climatology proposed by Kinne et al., 2013 (doi:10.1002/jame.20035.), which exhibit typical mean values around 0.12 in the SEVIRI disk.

As concern the underestimation at large AOT, very high AOT normally correspond

to local events, especially in Europe (e.g. plume, fire), therefore the AOT obtained by the retrieval from the satellite pixel containing the AERONET station will be lower than the one measured by the AERONET tower (Jiang et al., 2006, https://doi.org/10.1016/j.rse.2006.06.022). The processing of more data would be necessary to increase the number of points with large AOT. Regarding PROBA-V, since the spatial resolution is one km and it has a blue band, overestimation at low AOT should not be present in the data set as is the case for SEVIRI. The retrieval from PROBA-V observations is affected by additional problems: The poorer radiometric performances which decreases the importance of the information derived from the observations The lack of a thermal channel that leads to an unreliable cloud mask

We acknowledge that fact that there is an issue with these results that underperform AOT retrieval with respect to other algorithms retrieving AOT from other instruments. However we are not aware of any algorithm capable of delivering a good AOT product from PROBA-V over land surfaces. Within the PV-LAC project, the CISAR benefit compared to the current operational method has been proven (https://earth.esa.int/web/sppa/activities/instrument-characterization-studies/pv-lac-atmo/about).

Lines 402-409 of the revised paper read now:

"The GCOS requirements are a useful tool to compare different algorithms' performances. However, it should be considered that both SEVIRI and PROBA-V missions were not originally designed for AOT retrieval. GCOS requirement of 0.03 for low optical thickness translates into a radiometric noise requirement much better than 2 (1)% at 0.4 (0.6) $\mu$m, i.e., way below the radiometric performance of the SEVIRI and PROBA-V instruments (Table 3). The duration of the corresponding missions provides however a decisive advantage for the generation of AOT datasets from these instruments. In the following, the GCOS requirements are evaluated in terms of percentage of retrievals satisfying them."

[Figure]

Lines 410-414 of the revised paper read:

"This is in accordance with the poor radiometric performances of the polar orbiting instrument and with the outcome of the information content analysis performed in Sect. 4. The boxplots in Fig. 14 show an overestimation of the retrieval for low AOT and an underestimation for large AOT."

Lines 412-440 of the revised paper read now: "Additionally, very high AOT normally correspond to local events, especially in Europe e.g. plume, fire), therefore the AOT obtained by the retrieval from the satellite pixel containing the AERONET station will be lower than the one measured by the AERONET tower (Jiang et al., 2007). The histograms in Fig. 14 show that AOT values larger than 0.8 represent less than 5% of the total number AERONET observations, affecting the reliability of the statistics for high values of AOT. The processing of more data would be necessary to increase the confidence in results for high AOT values. Some examples of CISAR's ability to detect high AOT are shown in the Supplement. The overestimation of low AOT might originate from the different spatial scale between the satel- lite observations and the ground measurements. Most of the selected AERONET stations are located in Europe (Fig. 1), where the SEVIRI pixel resolution is about 5x8 km (as opposed to 3x3 km at the subsatellite point), which is compared to AERONET point measurement. The probability of residual cloud contamination at this scale might thus explain part of the overestimation (Henderson and Chylek (2005), Chand et al. (2012)). Furthermore, the shortest SEVIRI spectral band is centred at 0.67 $\mu$m, where the sensitivity to low optical thickness is about 2 times smaller than in the blue spectral region. Consequently, the retrieval in these cases essentially relies on the prior information regardless the very large associated uncertainty. Despite the presence of a blue band and a better spatial resolution (1 km), the retrievals from PROBA-V observations still show overestimation at low AOT, due to the poor radiometric performances which decrease the importance of the information derived from the observations and to the lack of a thermal channel that leads to an unreliable cloud mask."

16. Line 477-480. I don't understand these comments about Fig. 17. CISAR/SEVIRI is in very good agreement? As CISAR/PROBA-V, it misses the peak of the distribution. Also, CISAR/PROBA-V is said to be underestimating the fraction but so does CISAR/SEVIRI. The significance of the ratio should also be discussed. What are typical ranges?

"Very good agreement" has been removed and replaced by "It can been seen that the distribution related to CISAR retrievals from SEVIRI and PROBA-V observations underestimate the fine mode concentration for $\tau F/\tau C >3$.". I'm not sure I understand if the reviewer is referring to typical ranges of the fine/coarse mode ratio. In this case, the AERONET data can be taken as reference.

17. The relative magnitude of those"spikes" in Figs. 19 and 20 are worrisome. For the causes you attribute, shouldn't they confirm that your choice of the three vertices is inadequate?

The aerosol vertices have been adjusted as suggested by the reviewer. With the new vertices the magnitude of the spikes strongly decreases. The percentage of points falling on these values is reported in Table 1. The percentages in Table 1 are in agreement with the solution space encompassing about the 80% of the AERONET data.

Table 1 Percentage of SSA and Asymmetry factor retrievals falling on the spikes in Fig. 17 and 18 w0 g $0.6\mu$m $0.8\mu$m $0.6\mu$m $0.8\mu$m SEVIRI 20% 23% 8% 7% PROBA-V 15% 31% 5% 4%

18. Line 487: I take the chance here to expand on previous comments. "Coarse mode characterization" is very far-fetched. The algorithm is not so much retrieving surface and aerosol properties, as much as two aerosol radiative properties and a set of RPV parameters white variability has not been ascertained. Even here, you've already got problems with unreliable retrievals of fine-to-coarse ratio, so much that you focus on the ratio being less or larger than 1. For these reasons, the title sounds a bit pretentious and should be adjusted accordingly. Omega and g are properties but based on the

current title nowadays most readers would expect an extended set of microphysical and optical properties

Indeed, CISAR retrieves the Single Scattering Albedo and the phase function for the aerosols and the RPV parameters for the surface. As described in Part I, each of the surface parameters controls the BRF differently, describing its magnitude, shape, anisotropy and hot spot. Any previously present reference to micro-physical aerosol properties was erroneous and has been removed. The title is therefore consistent with what the algorithm retrieves.

19. Sec. 6.3: how about Carpentras?

The timeseries is shown in Fig. 3 where the MODIS data have been filtered according to their associated quality flag (https://lpdaac.usgs.gov/sites/default/files/public/modis/docs/MODIS_LP_BRDF+Albedo_QA_Tutorial-4.pdf). It can be seen that the MODIS timeseries shows some issues and cannot be considered reliable. This might also partially explain the scattering in the BHR density plots in the supplements. Using MODIS to simulate satellite observations in the attempt of proving the FASTRE capability of correctly characterise the satellite observations we underestimated the effort to collect ground truth RPV parameters.

20. Line 545-547: This is either too obvious or a concept I don't get. You don't describe state variables, you retrieve them, so isn't just that the algorithm fails?

Following comment 3, this sentence has been removed.

21. The manuscript should report complete statistics on the number of analyzed scenes, so that the retrievals can be put in context. I'm not sure this is what happens in Table 11.

The concept of "report complete statistics on the number of analyzed scenes" is not clear. Unfortunately, we cannot answer this comment.

[Figure]

Please also note the supplement to this comment:
https://www.atmos-meas-tech-discuss.net/amt-2018-265/amt-2018-265-AC1-supplement.pdf

[Figure]

[Figure]

Counts

−0.10    −0.05    0.00    0.05    0.10    0.15    0.20

Jacobians

**Fig. 1.** Merged Figure 6

[Figure]

**Fig. 2.** Bias between CISAR retrieved AOT from SEVIRI (blue) and PROBA-V (red) and AERONET data. The histograms show the distribution of the AERONET data.

**Fig. 3.** BHR timeseries at 0.6um over Carpentras

**Supplement:**

This is just a preview and not the published paper.

[revised manuscript text omitted]

This is just a preview and not the published paper.

[Figure]

[Figure]

(a) SEVIRI

(b) PROBA-V

Figure 20: CISAR retrieved BHR (blue dots) from SEVIRI (upper panel) and PROBA-V (lower panel) and MODIS Land Product (yellow triangle) averaged on SEVIRI and PROBA-V pixels over Zinder Airport (Niger, Africa). The results are shown for the sensors band centred at 0.6 $\mu m$ over 2015. The vertical bars represent the CISAR retrieval uncertainty.

CISAR retrievals and the MODIS Land Product dataset show similar seasonal trends. Figure 20 shows the BHR timeseries over Zinder Airport (Niger, Africa), as retrieved from the CISAR algo-

This is just a preview and not the published paper.

[Figure]

[Figure]

525 rithm applied to SEVIRI and PROBA-V observations and from MODIS Land Product. The rainy
season, going from May 20 to October 5 (Weatherspark.com, 2018), is distinguishable in all the
datasets, although CISAR resolves a larger seasonal variation with respect to MODIS product. The
effect of the updating mechanism on the surface prior described in Section 2.4 is also visible, espe-
cially in Fig. 20b, where the retrieval uncertainty decreases in time, as the prior information on the
530 surface is better defined.

**7   Discussion and conclusion**

This paper describes and evaluates the CISAR algorithm  to satellite observations ac-
quired from geostationary and polar orbiting instruments. The theoretical aspects of CISAR, a new
generic algorithm for the joint retrieval of surface reflectance and aerosol properties, with continuous
535 variation of all the state variables in the solution space, are described in Part 1.  CISAR is
applied to simulated noise free observations in the principal plane. This paper provides an evaluation
of the algorithm in non ideal situations, *i.e.*,  actual satellite observations acquired
from  SEVIRI and PROBA-V.

The proposed retrieval method relies on an OE approach which consists in the inversion of FAS-
540 TRE, a simple radiative transfer model composed of two horizontal layers. The first step of CISAR
algorithm evaluation consists thus in the evaluation of the inverted forward model (Section 2.5). The
FASTRE model is accurate within 1% to 3% percent when compared to a complex 1D radiative
transfer model. Higher uncertainties are observed in spectral bands affected by water vapour as a
result of the limited vertical discretisation. The FASTRE model capabilities to simulate actual ob-
545 servations has also been evaluated but revealed relative bias larger than 5%. This poor performance
partially results from the lack of accurate description of the state variables at the moment of the
satellite overpasses.

Despite the PROBA-V instrument has one blue channel which is not present on SEVIRI, the frequent
550 revisit rate of the latter provides more information for the retrieval of surface reflectance and aerosol

Table 11: CISAR retrieved BHR from actual observations comparison with MODIS in all the pro-
cessed bands.

|  | SEVIRI | | | PROBA-V | | | |
|---|---|---|---|---|---|---|---|
|  | 0.6 $\mu$m | 0.8 $\mu$m | 1.6 $\mu$m | 0.6 $\mu$m | 0.4 $\mu$m | 0.8 $\mu$m | 1.6 $\mu$m |
| Number of points |  | 7409 |  |  | 744 |  |  |
| Correlation | 0.925 | 0.820 | 0.860 | 0.763 | 0.891 | 0.774 | 0.890 |
| Root Mean Square Error | 0.045 | 0.067 | 0.080 | 0.030 | 0.051 | 0.092 | 0.085 |
| Mean Absolute Bias | 0.038 | 0.053 | 0.067 | 0.026 | 0.042 | 0.064 | 0.068 |

This is just a preview and not the published paper.

[Figure]

[Figure]

properties .

[revised manuscript text omitted]

---

## Author Comment (AC2) · 5 Oct 2018

1. My issue is that I see no evidence that this algorithm currently produces acceptable results. Fig. 14 is not good. It's not bad enough to imply your technique is without merit, but if that's the only plot you're going to provide, you will struggle to attract interest

in this algorithm as your correlation, bias and RMSE are worse than most products I've encountered. At the very least, you need to find some circumstances where your retrieval's ability to mix aerosol types produces a better retrieval than a more developed product (e.g. MODIS collection 6.1 or the Swansea University product from Aerosol CCI). Maybe biomass burning emissions from Africa or the industrial regions of China?

Please refer to the reply to Reviewer #1 for Fig. 14.

2. Also, the heritage of the algorithm and the plots in the supplement imply this method is a much better retrieval of the surface than of aerosol. I would warm to the paper more if it was arguing that you made a slightly better aerosol retrieval without harming the surface product rather than the current structure, which implies you were trying to make an aerosol retrieval and skims over the significant limitations in your current results.

Results on the surface BHR are now shown in Section 6.1 to present them prior the AOT. The following lines (506-511) are added to the discussion "The CISAR surface albedo is compared with the MODIS product, showing a correlation higher than 0.74 in all processed bands (to the exception of the NIR PROBA-V band). The better performances of CISAR in retrieving the surface reflectance rather than the AOT are explained by the larger contribution to the TOA BRF at the satellite of the surface. The little variance of the surface reflectance on a short time scale allows a good prior definition based on the previous CISAR retrievals."

3. §4 Though I'm pleased to see a discussion of information content in an atmospheric science paper, yours is rather unusual. You're using the magnitude of the Jacobian to argue which terms are the most important. However, the Jacobian has units and so the magnitude of different terms isn't direct comparable. To illustrate, consider Fig. 5, which you use to argue that 0 is a more dominant driver of changes in TOA radiance than $\theta$. A small change in surface reflectance could be of order $10-3$, which would produce a change of about $10-3$ in y (as the Jacobian is approximately unity). A small

change in viewing angle could be 1 âŮę and, if the Jacobian shown was in units of degrees, that would imply a change of -0.2 in y, which is much larger than that for 0. (The change is still larger if the units are radians.) The value of the Jacobian must be scaled by an appropriately small change to be compared to other values. Optimal estimation already has a mechanism to evaluate this. It's called the averaging kernel and Eq. 2.78 of Rodgers (2000) defines it as, A = KTS −1 K + S −1 a −1 KTS −1 K. You likely already calculate this when determining the entropy (see Eq. 2.80). A row of the averaging kernel summarises the contributions of each state vector element to the retrieval of each other variable while the diagonal elements quantify the reliance on the prior. (Things are slightly complicated by the addition of smoothing, H, terms to your cost function. The difference is subtle; ask Oleg Dubovik about it.) For your retrieval, I would expect the diagonal of A for 0 to be close to one and k to be closer to zero. It would also illustrate the interdependence of the different terms. I don't know if the average reader would find such an analysis easier to understand. Averaging kernels, though very powerful, are confusing. I tend to put them in supplementary material for people that care to find. If you don't switch to averaging kernels, label your plot axes as derivatives rather than Jacobians (e.g. the x-axes on Fig. 6 is dy d$\tau$ ) so readers have some chance of understanding what's being plotted. More practically, I'd say a superior test to use in §5.2.4 would be the number of degrees of freedom for noise (e.g. n − tr A)

Thanks for your suggestions. The analysis of the information content is now performed on the Jacobians scaled on the variability range of each variable, to account for the different units. Figure 5 now shows the scaled Jacobians, and the axes are labelled accordingly. We prefer not to switch to averaging kernels as they are confusing, as explained by the reviewer.

4. §4 More generally, I'm not sure why this section is so long. It's worthwhile to point out that the retrieval's sensitivity is a function of what is observed, but there must be a more efficient way to show that the retrieval has minimal sensitivity at some times of

day/year.

It is very important to discuss the challenges associated with retrieving information from satellite observations and the difficulty to get a retrieval with constant retrieval with time as the magnitude and sign of the Jacobians can change.

5. Tab. 2 This is a substantial problem. You should be more upfront about the current limitations of your method and outline in more detail what you intend to do about them. There's nothing wrong with incremental progress. This also affects L568.

The FASTRE validation is now presented in a different way. The comparison between simulations and actual observations has been removed. Now FASTRE is only evaluated against a much more accurate radiative transfer model (RTMOM) is SEVIRI and PROBA-V bands, as introduced in Part I. Please refer to the answer to the comment #3 and #19 of Reviewer #1.

6. Fig. 4 I agree with the other reviewer in wondering why you selected vertices that exclude a significant population of observed aerosols.

The aerosol vertices have been adjusted in order to encompass a wider area and include the peak of the distribution.

7. L299 I strongly suspect that there is less information content in the polar data because you ascribed more uncertainty to it ($\sigma c$ and $\sigma\theta$), not because of anything intrinsically advantageous to the geostationary view. This affects your conclusions on L555 and L561. (My opinion is that geostationary data is superior when you need temporal resolution and polar data superior when you need global coverage.)

The reviewer is right, given the larger radiometric uncertainty, PROBA-V data carry less information than SEVIRI ones. Lines 262-264 (previously 297) now read: "The distribution of the surface and AOT entropy related to SEVIRI observations exhibits higher values compared to the one related to PROBA-V observations, given the larger radiometric uncertainty associated to the observations acquired by the polar orbiting

satellite.". Lines 306-310 are eliminated. Lines 549-551 (now 491-493) have been changed to: "Though the PROBA-V instrument has one blue channel which is not present on SEVIRI, the better radiometric performances of the geostationary satellite provide more information for the retrieval of surface reflectance and aerosol properties than the polar orbiting instrument.". Line 561 (currently 501-502) reads now "These differences are explained by the different information content associated to the observations acquired by the two satellites".

8. L321 Do you mean that the magnitude of the cost increases with the number of observations because there are, well, more observations? – L297 of Part 1 addressed something similar by putting a scaling into the cost function; you could do that. – The cost function is (theoretically) a $\chi$ 2 distribution with a number of degrees of freedom equal to the number of observations. Using that model, the cost can be converted into a probability that the fit is coincidental and a threshold for retrieval quality defined in terms of that (for example, keeping only retrievals with less than a 5 % probability of being the result of chance). – Regardless, I agree that filtering by cost alone will not identify retrievals with minimal sensitivity.

Thanks for your comment. Indeed, the cost function could be converted in some form of probability and used in the quality indicator computation. However, this test would be performed on the entire accumulation period rather than on a single observation. In CISAR a different QI for each observation is computed to proceed as in test 3.

9. §5 This section is very difficult to follow and needs redrafting with help from someone unfamiliar with the method. Switching between p, q, and QI doesn't help, especially when 1 is a good value for one while 0 is a good value for the other. It would be substantially easier to follow if you provided a decision tree.

The whole section has been rewritten. The term q has been removed. The terms pi now represent the different tests. Section 5.2 now incorporates part of Sect. 5.3, leaving the latter much simpler. Good values are associated with 1, bad values with 0.

It was already the case, but probably it was not very clear. Line 204 reads "Each test pi can assume values 305 between 0 (bad quality) and 1 (good quality)." and line 369 reads "The final QI(ti) ranges from 0 to 1, where 0 designate a poor quality retrieval and 1 indicates a reliable solution.". Please refer also to comment #8 of reviewer #1.

10. §5.2.3 Though I understand the motivation behind this test, I should point out that $ym−y0$ $\sigma0$ is normally distributed. As such, 31.8 % of observations would be expected to fail your test by simple chance.

The reviewer is indeed right. The choice of this test and the relative thresholds derives from the choice of being more or less conservative.

11. L360 I agree with the other reviewer that the lack of discussion of a cloud masking is surprising. PROBA-V lacks thermal channels, making it difficult, but you have no problems on SEVIRI.

"Cloud contamination" has been replaced by "cloud mask omission errors". An external cloud mask is applied (Sect. 2.3), however some clouds might not be detected and lead to the overestimation of the AOT.

12. L425 This extra test should have been mentioned back in §5.2.5. More justification of this work around is necessary.

This extra test is now mentioned in Sect. 5.2 and the following sentences (lines 362-364) are added: "Low entropy might be due to a reliable prior information, with a low associated uncertainty. Similarly, the uncertainty reduction would be very large in case of prior information with a very large uncertainty on the state variable."

13. L453 A factor of two is not a 'slight' overestimation and the fact that your retrieval was this bad eight years ago does not forgive it's failure now.

Please see answer to comment #15 of Reviewer #1.

14. L478 That isn't good agreement. A good agreement can be seen between the red

and green lines in Fig. 18(a).

The comment on Fig. 17 (now Fig. 15) reads now (lines 443-445): "It can been seen that the distribution related to CISAR retrievals from SEVIRI and PROBA-V observations seem to underestimate the fine mode concentration for $\tau F/\tau C > 3$."

15. §6.2 These comparison look good! Why not give us a version of Fig. 14 for SSA and g? Considering they're what you retrieve, I wouldn't be surprised if you could estimate them better than you could AOT. Wouldn't make me think the product was any better as most users want AOT, but they aren't many global SSA and if you could provide one, even if it's very uncertain, that would be something worth writing about.

The correlation is strongly dependents on the amount of variability in the datasets (Goodwin et al., 2006 https://pdfs.semanticscholar.org/b6cf/001cbab0375a96c370585462dd3c163669af.pdf). As the variability range of the aerosol single scattering properties is very limited (about 10%), we don't find it useful to show the same kind of plot as Fig. 14.

16. L116 There are many potential calibration methods for SEVIRI. If you're using IMPF or GSICS, could that be mentioned explicitly? If you're using something in-house, a citation would be appreciated.

The calibration method used within this study is the one proposed by Govaerts et al. (2013), as specified in Sect. 2.2. GSICS provides routinely correction factors from IMPF values only for the thermal channels, not for the solar ones.

17. L145 Why make this approximation? Is the calculation of the other terms computationally expensive?

Yes, the calculation of the other term is computationally expensive as it implies the calculation of additional partial derivatives.

18. Eq.6 This seems a strange choice. Why not the standard deviation or interquartile range or a constant value based on climatology?

The range in which they vary is less conservative than the standard deviation or the interquartile. We don't want to impose a too strong prior. We are not using any climatology for the surface and we do not intend to.

19. §2.4(1) What's the value of Nmin? Why increase the uncertainty by 5 % per day rather than any other amount?

Nmin has been added is Table 8. The prior uncertainty is increased by the arbitrary value of 5% per day in order not to rely on a solution retrieved too far away in time from the current inversion. This value has been empirically adjusted.

20. §2.5 I'd actually prefer to see a thorough sensitivity study of bias as a function of the various parameters rather than the simple 1 - 3 % uncertainty you've added, but that can be in a third paper.

Thanks for the suggestion, we might consider this for a future study. For the time being it has been implemented in this way for efficiency purposes.

21. P12L2 In my experience, the first guess is set to reduce the number of iterations needed to reach a solution. Avoiding local minima involves checking the shape of state space around the final solution or annealing (i.e. running multiple retrievals on the same data).

Indeed, one alternative solution to avoid local minima is to run multiple retrievals on the same data. However, this is also computationally expensive. The idea behind alternated first guess is to simulate the annealing running the inversion starting from different first guess for each observation, rather than repeating N times the same inversion.

22. Eq.8 So you're using a different first guess for even and odd numbered time steps? That's peculiar and, on its own, I don't see how it avoids local minima.

The first guess of the RPV parameters is not defined to minimize the probability of falling into a local minima as for the AOT. As empirical results showed that even a

slight overestimation or underestimation of the surface can lead to larger bias in the AOT retrieval, the different first guess is set to not get stuck in a over/under-estimation situation.

23. §4 The third paragraph covers four pages. Perhaps split it up.

This has been done.

24. P17L1 As the sensitivity drops through the day, I would expect the uncertainty to increase.

The AOT retrieval uncertainty depends not only on the Jacobians, but also on the temporal and spectral smoothness constraints and the quality of the surface. However Fig. 1 shows that for high Jacobians the retrieval uncertainty decreases.

25. L351What is the maximum number of iterations?

It is 20, this has been added to Table 8.

26. L352 Could you clearly state that p0 = 1 in all other circumstances. I wasn't certain of that till I got to Eq. 15.

This is now clearly stated. Line 352 (now 311) reads "When the maximum number of iteration is reached p0 is equal to 0, otherwise p0 = 1."

27. §5.2.2 Did you ever explore using the a priori cost for this test (i.e. the difference between the retrieval and the prior)?

Thanks for the suggestion. We might explore this option in the future.

28. Eq. 11 Aren't the y terms vectors? If so, wouldn't this require some sort of sum?

The formula is now more explicitly written. As I'm actually considering the maximum mismatch among the different bands within this test (this was not explicitly written earlier), those terms are vector components.

29. L371 Didn't you have to calculate the full Jacobian to perform your inversion? I see

your point, but this is a lot of explanation for why you don't use something you should already have

I do have the full Jacobians, but considering them would require even more tests and manipulations. Anyways, as suggested from Reviewer #1, this part has been shortened

30. L379 I assume that if I ask for a justification of this statement, I will be told to go look at your papers from 2010 so I will make this sarcastic remark instead.

Indeed, in Wagner et al. (2010) the impact of the surface prior update on the covariance matrix is analysed. Furthermore, in Luffarelli et al. (2017) the effect of the updating mechanism on the retrieval is also analysed (https://ieeexplore.ieee.org/document/8035227).

31. Eq.15 For the sake of future readers' comprehension, please restrict qi to the range [0, 1] and make QI a simple product rather than use the difficult to comprehend $1 -$ max(q, 1) construction.

The range of qi has been restricted to [0,1]. However the QI construction cannot be replaced by a simple product as it would give the same results.

32. L409 Please specify this sigmoid function (or at least give it's width).

The width of the sigmoid function is now specified in lines 333-335 "When the mismatch assumes values within the range defined by T1 and T2, thresholds excluded, a value between a minimum m and 1 is assigned to the test 3 through a sigmoid function with width equal to $10/(T2 -T1)$ (Fig. 11)."

33. Fig. 14 Can we please have a version of this plot as a 2D histogram in the supplement, similar to the ones already there for the BRF?

We are not sure what the reviewer refers to, as there are no histograms here for the BRF.

34. The y-axis of Figs. 5, 6, 9, 10, 15, 17, 18, 19 should probably be 'Fractional counts'

considering they clearly have non-integer steps.

This has been done.

35. Fig.16 (b) and (c) aren't that interesting or helpful. Perhaps make (c) an inset in (a).

The figure has been removed.

All grammatical suggestions have been implemented.

Please also note the supplement to this comment:
https://www.atmos-meas-tech-discuss.net/amt-2018-265/amt-2018-265-AC2-supplement.pdf

———————————————————

[Figure]

[Figure]

**Fig. 1.** AOT retrieval uncertainty as a function of the Jacobians

**Supplement:**

[revised manuscript text omitted]

---

## Referee Report (RR1)

**Re-review of 'Joint retrieval of surface reflectance and aerosol properties with continuous variations of the state variables in the solution space: part 2: Application to geostationary and polar-orbiting satellite observations'**

The paper has been much improved by the author's revisions, especially §5. I'm still unconvinced the algorithm is producing useful data, but I'm not opposed to the work's publication after some minor corrections.

My primary comments concern your response our opinions of Fig. 14.

- *The bias between the CISAR retrieval and the AERONET data is shown in Fig. 2, which shows different performances for SEVIRI and PROBA-V. These differences show that the bias does not only depend on the CISAR algorithm itself, but also on the quality of the processed data.*

  Your SEVIRI and PROBA-V implementations use a different number of channels, so it is entirely reasonable for them to exhibit different bias profiles. This does not excuse the fact that your algorithm exhibits biases of around 50% in your own validation (and an independent validation would be expected to find worse comparisons). I appreciate that the two satellites you use don't exhibit the radiometric quality of MODIS or AATSR. That is expressed (as I would expect) through the large uncertainty in your products relative to MODIS.

  However, Fig. 2 of your response only worsens my opinion of your results. A calibration offset should result in a retrieval bias that is (roughly) independent of optical depth. Lower quality detectors should give a wider scatter (which, admittedly, you appear to have). Cloud contamination should result in a positive bias. Yes, your results for SEVIRI are within the GCOS requirements for a particular range but your biases for AOD < 0.2 are almost 100%. As AOD is log-normally distributed, this region carries significant weight.

  I think we see Fig. 2 very differently. I expect you see the SEVIRI bias as a straight line around zero, that drops off above 0.7 due to the small volume of data and cloud contamination. I look that that line and, neglecting the last point, see a linear downwards trend. PROBA-V does the same, but with a different gradient. I can live with data with a large RMS — average could reduce it. I can live with data that has a bias — one can subtract it. Your data has a slope. I'd need both coefficients of $ax + b$ to bias correct your data and that's difficult. To my mind, an algorithm that sruggles to retrieve both small and large AOD provides little of use.

  So what do I think should be done? It would be inhuman to ask you to rubbish your own data in your own paper. Your revisions do better represent the quality of this data, but you primarily blame the instruments and cloud. If those were the only problem, you should resubmit the paper after applying it to a better instrument.

I continue to believe that your algorithm is conceptually interesting. What I need is a discussion of what you intend to do next. What aspects of the algorithm are you working on? Where do you think the problem lies?

- *It can be seen there only few points correspond to AOT > 0.8 (less than 5% of the total number of observations), affecting the reliability of the statistics for high values of AOT. The histograms have been added in Fig. 14.*

  Though I agree that large AOD events are rare, they can be very important. Large dust plumes seed the equatorial ocean, large fires impact air quality over entire continents, and volcanic eruptions affect international air travel. Most algorithms have trouble with high AOD, where the fundamental assumptions of such retrievals begin to break down, but they are an area of active development.

- *The overestimation rapidly decreases as the AOT approaches values of about 0.2.*

  This is unimportant as the line has to cross the axis somewhere.

- *We are not aware of any algorithm capable of delivering a good AOT product from PROBA-V over land surfaces.*

  I'm not aware of anyone having tried as I'd never heard of the instrument before reading this paper.

- With regard to my own point 33, your Fig. 14 shows the comparison of CISAR to AERONET through boxplots. In the supplement, you show a comparison of CISAR to MODIS through 2D histograms. I vastly prefer the later form of plot and was requesting a second version of Fig. 14 in the style of Fig. S1.

A few other thoughts:

- I'm not overly happy with the assumptions you make in §2.4 but they're rational. I will note, though, that on L148 you state you set the uncertainty to a 'high arbitrary value'. Unity is not high. When I wish to avoid setting a prior for a variable, I use a prior uncertainty of $10^8$.

- You reference an unusual number of reports and conference presentations. I know private companies struggle to justify publication costs but I'm slightly concerned that many background details for this algorithm haven't been peer reviewed.

- I think I now understand §5. I'm not fond of this manner of qualitative quality filtering, but it is commonly used so I'll not comment further.

- Is Fig. 12 actually binned (i.e. showing the average correlation for all retrievals with QI in a certain range)? If so, it would be useful to represent those ranges on the plot (e.g. the step histograms of matplotlib).

The English quality is decent. I include some corrections in the text below.

[revised manuscript text omitted]

---

## Author Response (AR2)

RE-REVIEW OF 'JOINT RETRIEVAL OF SURFACE REFLECTANCE AND AEROSOL PROPERTIES WITH CONTINUOUS VARIATIONS OF THE STATE VARIABLES IN THE SOLUTION SPACE: PART 2: APPLICATION TO GEOSTATIONARY AND POLAR-ORBITING SATELLITE OBSERVATIONS'

My primary comments concern your response our opinions of Fig. 14.

- *"The bias between the CISAR retrieval and the AERONET data is shown in Fig. 2, which shows different performances for SEVIRI and PROBA-V. These differences show that the bias does not only depend on the CISAR algorithm itself, but also on the quality of the processed data."*

Your SEVIRI and PROBA-V implementations use a different number of channels, so it is entirely reasonable for them to exhibit different bias profiles. This does not excuse the fact that your algorithm exhibits biases of around 50% in your own validation (and an independent validation would be expected to find worse comparisons). I appreciate that the two satellites you use don't exhibit the radiometric quality of MODIS or AATSR. That is expressed (as I would expect) through the large uncertainty in your products relative to MODIS.

The PV-LAC validation report (https://earth.esa.int/documents/700255/2632405/PV-LAC_ATMO_VR_v2.2.pdf/4c46403b-bfe5-4208-bfdf-2d42585d6589), prepared by Erwin Wolters from VITO (De Vlaamse Instelling voor Technologisch Onderzoek) through an independent verification, does not show much worse results, keeping in mind that the algorithm improved since then.

However, Fig. 2 of your response only worsens my opinion of your results. A calibration offset should result in a retrieval bias that is (roughly) independent of optical depth. Lower quality detectors should give a wider scatter (which, admittedly, you appear to have). Cloud contamination should result in a positive bias. Yes, your results for SEVIRI are within the GCOS requirements for a particular range but your biases for AOD < 0.2 are almost 100%. As AOD is log-normally distributed, this region carries significant weight.

We recognize the limitations of CISAR retrieval for low AODs when applied on SEVIRI and PROBA-V data. However, the bias for AOD = 0.1 is 54% for SEVIRI and 48% for PROBA-V. For AOD=0.15 the bias decreases to 18% for SEVIRI and 27% for PROBA-V.

I think we see Fig. 2 very differently. I expect you see the SEVIRI bias as a straight line around zero, that drops off above 0.7 due to the small volume of data and cloud contamination. I look that that line and, neglecting the last point, see a linear downwards trend. PROBA-V does the same, but with a different gradient. I can live with data with a large RMS—average could reduce it. I can live with data that has a bias — one can subtract it. Your data has a slope. I'd need both coefficients of ax+b to bias correct your data and that's difficult. To my mind, an algorithm that sruggles to retrieve both small and large AOD provides little of use.

So what do I think should be done? It would be inhuman to ask you to rubbish your own data in your own paper. Your revisions do better represent the quality of this data, but

you primarily blame the instruments and cloud. If those were the only problem, you should resubmit the paper after applying it to a better instrument.
I continue to believe that your algorithm is conceptually interesting. What I need is a discussion of what you intend to do next. What aspects of the algorithm are you working on? Where do you think the problem lies?

The sentence "The cloud mask omission errors impact on the AOT overestimation at low optical thickness deserve additional work." has been moved at line 516 and the following lines have been added afterwards:
"In order to reduce the impact of cloud contamination in the AOT retrieval, a new version of the CISAR algorithm is under development in the framework of the ESA-SEOM ConsIstent Retrieval of Cloud Aerosol Surface (CIRCAS) project (www.circas.eu). The new version of CISAR aims to retrieve both the AOT and the Cloud Optical Thickness (COT), overcoming the need of an external cloud mask. Within the CIRCAS project CISAR will be applied to observations acquired by the Sea and Land Surface Temperature Radiometer (SLSTR) on-board Sentinel-3."

- *It can be seen there only few points correspond to AOT È 0.8 (less than 5% of the total number of observations), affecting the reliability of the statistics for high values of AOT. The histograms have been added in Fig. 14.*

   Though I agree that large AOD events are rare, they can be very important. Large dust plumes seed the equatorial ocean, large fires impact air quality over entire continents, and volcanic eruptions affect international air travel. Most algorithms have trouble with high AOD, where the fundamental assumptions of such retrievals begin to break down, but they are an area of active development.

   As stated in the manuscript, we believe that the processing of more data would be necessary to increase the confidence in the results for high AOT values.

- *The overestimation rapidly decreases as the AOT approaches values of about 0.2.*
   This is unimportant as the line has to cross the axis somewhere.

   It is not clear to me what the reviewer means by this.

- *We are not aware of any algorithm capable of delivering a good AOT product from PROBA-V over land surfaces.*
   I'm not aware of anyone having tried as I'd never heard of the instrument before reading this paper.

   The operational product includes an atmospheric correction method. However, when compared with CISAR retrieval during the PV-LAC project, it did not show promising results.

- With regard to my own point 33, your Fig. 14 shows the comparison of CISAR to AERONET through boxplots. In the supplement, you show a comparison of CISAR to MODIS through 2D histograms. I vastly prefer the later form of plot and was requesting a second version of Fig. 14 in the style of Fig. S1.

  During the Aerosol-CCI project, it was strongly advised to show boxplots for the AOD retrieval, hence our choice.

A few other thoughts:
- I'm not overly happy with the assumptions you make in §2.4 but they're rational. I will note, though, that on L148 you state you set the uncertainty to a 'high arbitrary value'. Unity is not high. When I wish to avoid setting a prior for a variable, I use a prior uncertainty of $10^8$.

  Reviewer#1 seemed to believe the opposite. From his annotate pdf "2.0 for the coarse mode is unreasonably high". An uncertainty of 1.0 associated with values < 0.2 (this is the cases for the AOD climatology values) can be considered as a high value.

- You reference an unusual number of reports and conference presentations. I know private companies struggle to justify publication costs but I'm slightly concerned that many background details for this algorithm haven't been peer reviewed.

  We will be happy, in the future, to cite this paper and its companion.

- I think I now understand §5. I'm not fond of this manner of qualitative quality filtering, but it is commonly used so I'll not comment further.

- Is Fig. 12 actually binned (i.e. showing the average correlation for all retrievals with QI in a certain range)? If so, it would be useful to represent those ranges on the plot (e.g. the step histograms of matplotlib).

  The final QI is rounded to one decimal place; therefore no binning is performed in Fig. 12.

All the reviewer's suggestions in the pdf were implemented, with the exception of:
- "CISAR has been applied to SEVIRI and PROBA-V observations acquired over 20 AERONET stations". The reviewer suggestions was to replace "over" with "from". However, the satellite observations are not acquired from AERONET, but over an area surrounding the AERONET station.
- "PROBA-V satellite mission is intended to ensure the continuation of the Satellite Pour l'Observation de la Terre 5 (SPOT5) VEGETATION products since May 2014". The reviewer suggestion was to replace "since" with "begun in". However, it was preferred to replace "since" with "starting from".
- "FASTRE uncertainty is in the range of 1% - 3% (Table 6), which is smaller or equal to the instrument radiometric noise." The reviewer suggestion was to replace "smaller

or equal" with "equivalent". However, this would change the meaning of the sentence.

- The reviewer suggests eliminating the word "however" at L406. Nonetheless, it is needed to relate to the previous sentence and explaining why, despite the poor radiometric performances, the AOT retrieval from the two instruments is meaningful.

[revised manuscript text omitted]